# Poly(ADP-ribosyl)ating pathway regulates development from stem cell niche to longevity control

Guillaume Bordet[1], Elena Kotova[2], Alexei V Tulin[1]

The regulation of poly(ADP-ribose) polymerase, the enzyme responsible for the synthesis of homopolymer ADP-ribose chains on nuclear proteins, has been extensively studied over the last decades for its involvement in tumorigenesis processes. However, the regulation of poly(ADP-ribose) glycohydrolase (PARG), the enzyme responsible for removing this posttranslational modification, has attracted little attention. Here we identified that PARG activity is partly regulated by two phosphorylation sites, ph1 and ph2, in *Drosophila*. We showed that the disruption of these sites affects the germline stem-cells maintenance/differentiation balance as well as embryonic and larval development, but also the synchronization of egg production with the availability of a calorically sufficient food source. Moreover, these PARG phosphorylation sites play an essential role in the control of fly survivability from larvae to adults. We also showed that PARG is phosphorylated by casein kinase 2 and that this phosphorylation seems to protect PARG protein against degradation in vivo. Taken together, these results suggest that the regulation of PARG protein activity plays a crucial role in the control of several developmental processes.

## Introduction

Poly(ADP-ribose) polymerase 1 (PARP-1) uses NAD as a substrate to synthesize poly(ADP-ribose) polymer (pADPr) on the surface of nuclear proteins (1) (Fig 1A). The poly(ADP-ribosyl)ation pathway regulates many nuclear functions, including DNA repair, chromatin structure, and transcription initiation, as well as pre-mRNA fate, via alternative splicing (1, 2, 3, 4, 5, 6, 7, 8, 9, 10), by altering the physical and enzymatic properties of acceptor proteins, which, owing to the presence of poly(ADP-ribose) phosphate moieties, become highly negatively charged and thus dissociate from their target nucleic acids (1, 2) (Fig 1A). The automodification of PARP-1 causes its dissociation from chromatin and, thus, autoinactivaed (Fig 1B). Poly(ADP-ribosyl)ation, which normally adds from 2 to 200 ADP-ribose residues to a single site on the acceptor proteins and PARP-1

itself, is reversed by poly(ADP-ribose) glycohydrolase (PARG) that degrades poly(ADP-ribose) and thus removes pADPr from proteins (11, 12, 13) (Fig 1A and B). As a result, PARG can regulate the cycle of PARP-1 activity by stripping poly(ADP-ribose) from the enzyme and enriching its inactive pool, which has also been implicated in DNA compaction, nucleosome assembly and other non-catalytic chaperon-like activities of PARP-1 (14) (Fig 1B). Antagonistic effects of PARP-1 and PARG on pADPr are also reflected in their distinct intracellular localization (Fig 1C), which may also explain the timing of changes in poly(ADP-ribose) levels during the cell cycle. Whereas PARP-1 protein is associated with chromatin and is readily available to modify nuclear proteins, PARG is enriched in soluble fraction of nucleoplasm and can interact with the PARP-1 and poly(ADP-ribose) network only after chromatin opens up and PARP-1 dissociates from it (Fig 1C). The regulation of PARP-1 has been extensively studied during the last decade, especially for its role in initiation and progression of malignant tumors, leading to the development of PARP inhibitors for cancer treatment (15, 16, 17, 18). Recently, PARG has also been suggested as a potential target in cancer treatment (18, 19, 20, 21). However, little is known about its regulation. Several phospho-proteomic studies have reported that PARG proteins become phosphorylated in humans (22, 23, 24, 25, 26). In addition, the global organismal phospho-proteomics screen reveals heavy phosphorylation of *Drosophila* PARG at embryonic stages (27) (Table S1). However, the effects of phosphorylation on PARG function in mammals or *Drosophila* remain unclear.

*Drosophila melanogaster* is a good model to study pADPr regulation because the *Drosophila* genome only encodes a single PARG with a single splicing isoform (Fig S1A). *Drosophila parg* null mutation (*parg^27.1*) results in the animal's death at pupal stage, suggesting that PARG is essential for normal development (12). *Drosophila parg^27.1* mutants accumulate high quantities of intracellular pADPr (Fig 1D). The absence of functional PARG to hydrolyze pADPr leads to the dissociation of automodified PARP-1 from chromatin and its accumulation in Cajal bodies and, hence, the disruption of its function (Fig 1E) (12). Cajal bodies are spherical sub-organelles found in the nucleus of proliferative or metabolically active cells and are possible sites of assembly or modification of the transcription machinery of the nucleus (8). Although PARG protein functions are highly conserved among eukaryotes, the PARG

---

[1]University of North Dakota, Grand Forks, ND, USA   [2]Fox Chase Cancer Center, Philadelphia, PA, USA

Correspondence: Alexei.Tulin@und.edu

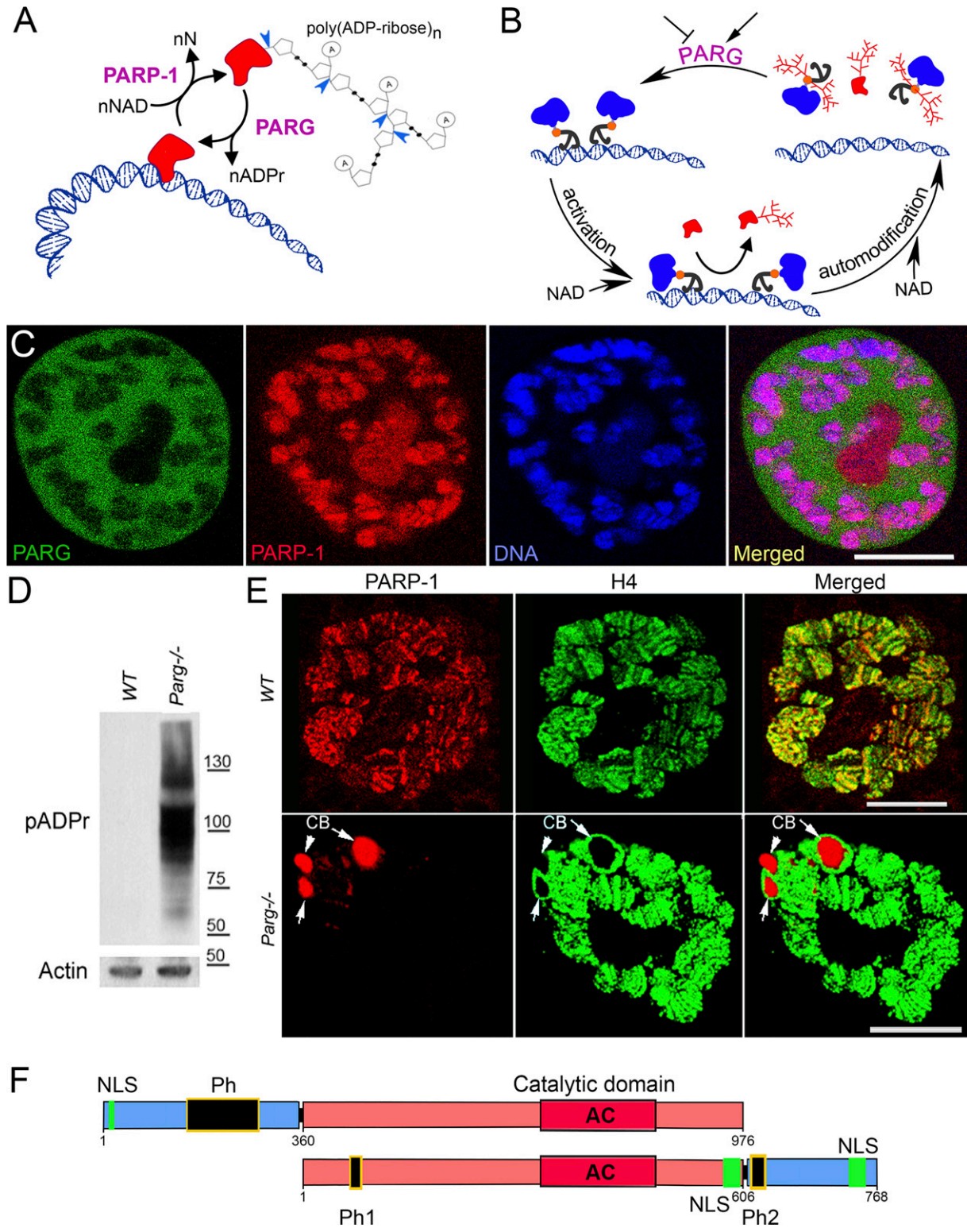

**Figure 1. poly(ADP-ribose) glycohydrolase (PARG) regulates poly(ADP-ribose) turnover in the cell.**
**(A)** poly(ADP-ribose) polymer (pADPr) turnover. Blue arrowheads indicate cleavage points of poly(ADP-ribose) by PARG. PARP-1, poly (ADP-ribose) polymerase 1; PARG, poly (ADP-ribose) glycohydrolase; NAD, nicotinamide adenine dinucleotide; N, nicotinamide. **(B)** PARG regulates PARP-1 activity cycle: (1) unmodified PARP-1 binds to chromatin as an inactive protein; (2) enzymatic activation of PARP-1 leads to poly(ADP-ribosyl)ation of target proteins; (3) automodification of PARP-1 causes its dissociation from chromatin; (4) PARG removes pADPr from PARP-1 and returns PARP-1 to chromatin. **(C)** PARP-1 and PARG exhibit antagonistic localization within the nucleus. This image represents a single nucleus of a polyploidy salivary gland cell. Green is PARG::YFP; red is PARP-1::DsRed, and blue is DNA. Scale bar, 15 μm. **(D)** PARG knockout leads to an irreversible accumulation of poly(ADP-ribosyl)ated proteins: Western blot analysis using anti-pADPr antibody. Actin is shown as a loading control.

protein sequence presents several differences between mammals and *Drosophila* groups. All mammalian PARGs have an N-terminal regulatory domain of 360 amino acids that is not present in *Drosophila* species (Figs 1F and S2). This N-terminal regulatory domain contains almost all the phosphorylation sites reported in Human (22). On the other hand, PARGs of *Drosophila* species contain a highly conserved C-terminal domain of 162 amino acids (Figs 1F, S1B, and S3). None of the PARG phosphorylation sites reported in Human is conserved in *Drosophila* species. In this study, we discovered the presence of six PARG phosphorylation sites that are conserved among *Drosophila* species. We show that the disruption of these sites affects PARG function, suggesting that they play an important role in the regulation of PARG activity.

# Results

### PARG protein is phosphorylated in *Drosophila*

A previous study revealed that *Drosophila* PARG was phosphorylated at embryonic stage (27) (Table S1). All phosphorylated epitopes are located at two sites (Figs 1F, 2A and B, and S1B). The first site, Ph1, is located at the N-terminal domain of PARG next to the sequence reported to be a mitochondrion transport signal in mammals (31). The second site, Ph2, is located in the insect-specific C-terminal domain next to putative *Drosophila* NLS (29) (Figs 2A and S1B). Both sites are conserved among the genomes of *Drosophila* species (Fig S3).

To study the roles of *Drosophila* PARG phosphorylation in the Ph1 and Ph2 sites, we created a PARG-SBP recombinant construct (Fig 2C) wherein a WT PARG protein was fused with streptavidin-binding protein epitope designed for purification of protein complexes (33). The expression of PARG-SBP completely rescues *parg²⁷·¹* mutant viability, suggesting that this recombinant PARG protein is fully functional. We have performed affinity purification of complexes containing PARG protein and using the streptavidin-binding protein tag (SBP-tag) approach coupled to protein identification using LC-MS/MS (32). Purification of PARG-SBP from third instar larval stages of *Drosophila* (*parg²⁷·¹; PARG-SBP*) resulted in no identification of other proteins, except PARG itself and fragments of degrading PARG (Fig 2D). However, in addition to PARG and fragments of PARG in the PARG-SBP pull-down assay, we have also identified a heavily phosphorylated form of PARG. Among all possible phosphorylated peptides, we found only peptides corresponding to ph1 and ph2 sites, which perfectly match those previously reported in *Drosophila* phospho-proteome studies, excepted for $T^{623}$ that we did not detect phosphorylated (27) (Fig 2B and Table S2). This confirms that modifications of ph1 and ph2 occur in *Drosophila*, both at embryonic stages and later in development.

To further study PARG phosphorylation, we created transgenic *Drosophila* stocks expressing WT and mutated PARG fused to YFP (Fig 3A): PARG$^{WT}$-YFP (WT); PARG$^{EA}$-YFP (catalytically inactive); PARG$^{SA}$-YFP (phosphorylation mutant); PARG$^{SE}$-YFP (phosphorylation mimicking). All constructs, except catalytically mutant PARG$^{EA}$, rescued *parg²⁷·¹* null mutant fly viability. Mutating phosphorylation domains ph1 and ph2 (PARG$^{SA}$) significantly increase the level of pADPr compared with PARG$^{WT}$ (Fig 3B). This increase is coupled with a significant decrease in PARG$^{SA}$-YFP protein level compared with PARG$^{WT}$-YFP level (Fig 3C). The phosphorylation mimicking form PARG$^{SE}$ does not exhibit any difference in the level of pADPr compared with WT (Fig 3B), whereas the protein level is significantly higher compared with WT (Fig 3C). Interestingly, the level of PARG mRNA is similar among PARG$^{WT}$, PARG$^{SE}$, and PARG$^{SA}$ (Fig S4), suggesting that the difference of protein level we observed results from a difference during the translation process or in protein stability. Furthermore, when we ran a Western blot for an extended time, the presence of two bands for PARG$^{WT}$ was revealed (Fig S5). One band showed PARG-YFP at the expected molecular weight, and one upper band was similar to the phosphorylated PARG band we observed with PARG-SBP (Fig 2D). PARG$^{SE}$ and PARG$^{SA}$ only exhibit the lower band, corresponding to unphosphorylated PARG and suggesting, in turn, that only PARG$^{WT}$ is phosphorylated in vivo.

All four recombinant PARG proteins are localized exclusively in the nuclei, predominantly in soluble nucleoplasm, and they are mostly excluded from nucleoli and chromatin (Fig 3D). This proves that the phosphorylation is not involved in regulating PARG protein localization. Similar to PARG$^{WT}$, the expression of PARG$^{SE}$ isoform completely restores (from the arrest in Cajal bodies) PARP-1 localization in chromatin in *parg²⁷·¹* mutants. Meanwhile, PARG$^{SA}$ rescues PARP-1 localization, but only partially, and the amount of PARP-1 in chromatin is severely reduced (Fig 3D). In contrast, PARP-1 protein level remains similar in PARG$^{SA}$ compared with PARG$^{WT}$ (Fig S6). Taken together, these results suggest that PARG is phosphorylated in *Drosophila* and that its phosphorylation is important for correct PARP localization.

### Phosphorylation of PARG protein regulates *Drosophila* ovary germline stem cell (GSC) differentiation process

Previously, we reported that PARG protein activity controls stem cell maintenance in the *Drosophila* GSC niche (7). PARG is essential for GSC anchoring by regulating pADPr-dependent DE-cadherin expression. Here, we tested whether PARG phosphorylation contributes to GSC regulation. In *Drosophila* ovary, the GSC niche is located at the very tip of a specialized organ called the germarium (Figs 4A and B and S7A–C). The GSC niche divides asymmetrically to generate a new GSC and a cystoblast (CB) that undergoes four rounds of incomplete mitosis to form a 16-cell cyst (34). Even after

---

**(E)** Mutating *parg* leads to redistribution of PARP-1 protein from chromatin to Cajal bodies. Single nucleus of larval salivary gland cell (polyploid tissue) is presented for each experiment. *WT*, wild type; *parg⁻ᐟ⁻*, *Parg* null mutant. Red is PARP-1::DsRed protein, and green is histone H4 protein tagged with GFP. Arrows show three Cajal bodies accumulating automodified PARP-1. Scale bar, 15 μm. **(F)** Domain structure of human and *Drosophila* PARG proteins. AC, active center; NLS, nuclear localization signal; NES, nuclear export signal; MTS, mitochondrion transport signal. Ph1 and Ph2 represent putative phosphorylation sites reported in this work. The phosphorylation domain (Ph) present in human PARG includes almost all the phosphorylation sites reported in human, but not conserved in *Drosophila*. Neither mammalian NLS (28) nor human phosphorylation sites (22) are conserved among *Drosophila* species. The 577–602 NLS has been previously reported (29), whereas 733–758 NLS was predicted with NLS-mapper (30).

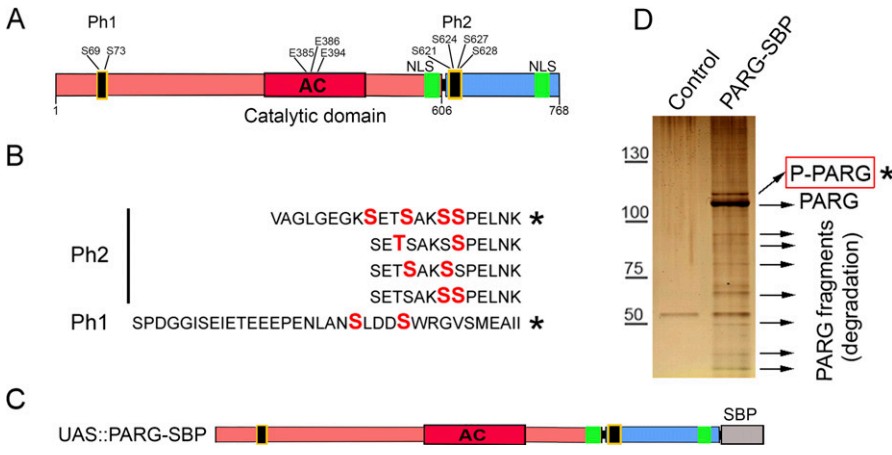

**Figure 2. poly(ADP-ribose) glycohydrolase (PARG) protein is phosphorylated in Drosophila.**
**(A)** PARG protein contains two phosphorylated regions (Ph1 and Ph2) located at the N- and C terminus of PARG polypeptide with two and four phosphorylated serine (S) residues (27). **(B)** The sequences of PARG phospho-peptides identified in *Drosophila* embryos (27). Asterisks label peptides confirmed in our lab for third instar larvae developmental stages. **(C)** Composition of the recombinant-transgenic PARG-SBP construct for in vivo experiments. Black boxes represent phosphorylation domains ph1 and ph2; red box represents catalytic domain, and green boxes represent NLS. **(D)** The phosphorylation of PARG was detected by mass spectrometry assay. The recombinant protein PARG-SBP was expressed in *parg* null *Drosophila*. Protein complexes were purified from larvae of this genotype (*parg^{27.1}*; *PARG-SBP*), along with *parg^{27.1}* control larvae. Proteins were separated using PAGE and detected using silver staining (the gel is shown). Bands corresponding to individual proteins were cut, and proteins were identified using LC-MS/MS (32). **(B)** Asterisk labels phosphorylated PARG, which contains peptides labeled on panel (B). See also Table S2.

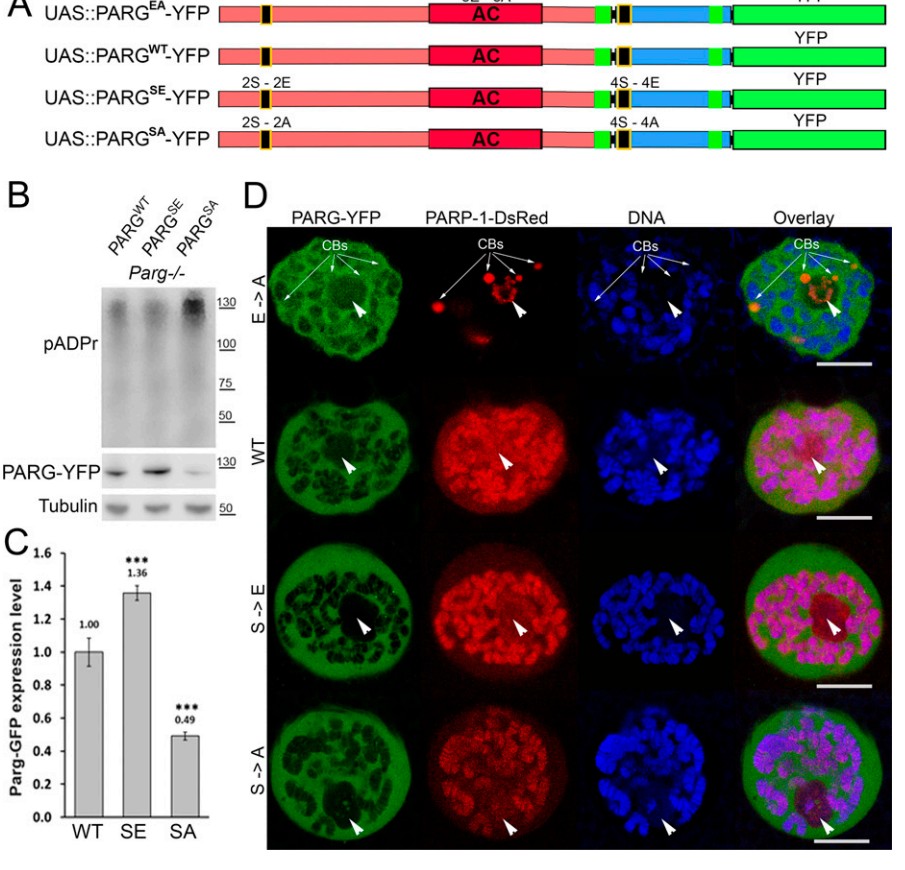

**Figure 3. poly(ADP-ribose) glycohydrolase (PARG) phosphorylation is essential for PARG function and PARG protein stability.**
**(A)** Composition of recombinant-transgenic PARG-YFP constructs for in vivo experiments. **(B)** Cellular pADPr level was assessed by Western blotting with anti-pADPr antibody. Total proteins were extracted from adult *Drosophila parg^{27.1}* mutants rescued with WT PARG^{WT}-YFP (WT), phospho-mutant PARG^{SA}-YFP (SA), and phospho-mimetic PARG^{SE}-YFP (SE). The blot was probed with anti-pADPr, anti-GFP, and anti-Tubulin antibodies for loading control. **(C)** Quantitation of relative band intensities (fold difference) shows that the level of phospho-mimetic PARG^{SE}-YFP (SE) protein is significantly higher than that of WT PARG^{WT}-YFP (WT) and that PARG^{SA}-YFP (SA) protein level is significantly lower than that of WT. **(B)** Calculation was performed on the basis of three independent experiments similar to those presented in panel (B). The statistical test used was a two-tailed *t* test. ***P-value < 0.01. **(D)** Dissected from live *parg^{27.1}*, larval salivary glands expressing full-length PARP-1-DsRed (Red) and PARG protein isoforms (green): WT; enzymatically inactive (EA), phosphorylation mimetic (SE), and phosphorylation mutant (SA) were stained with the DNA-binding dye Draq5 (blue) and analyzed by confocal microscopy for live imaging. A single nucleus is shown for each experiment. Positions of nucleoli are indicated with arrowheads. CB, Cajal body. Scale bar, 15 μm.

beginning their differentiation, CB and cysts can dedifferentiate into functional GSC (35). GSC and CB can be visualized by staining with an antibody against Hu li tai shao protein, the *Drosophila* homologue of adducin. This staining allowed the detection of a round-shaped organelle, one per cell, termed the spectrosome (Fig 4B), which is specific to both GSC and CBs (36). In WT ovary, each germarium contains only two GSCs on average (Fig 4C) (37). CB can be discriminated from GSCs by the expression Bag of marbles (Bam), a key component in the GSC differentiation process that is expressed in CB, but not in later stages of differentiation (37). PARG activity needs to be down-regulated in CBs (Fig 4B) to maintain the high level of pADPr required for pADP-ribosylation of hnRNP A1 to, in turn, inhibit DE-cadherin translation and release cells from the stem cell niche (7). These data show that PARG regulates the

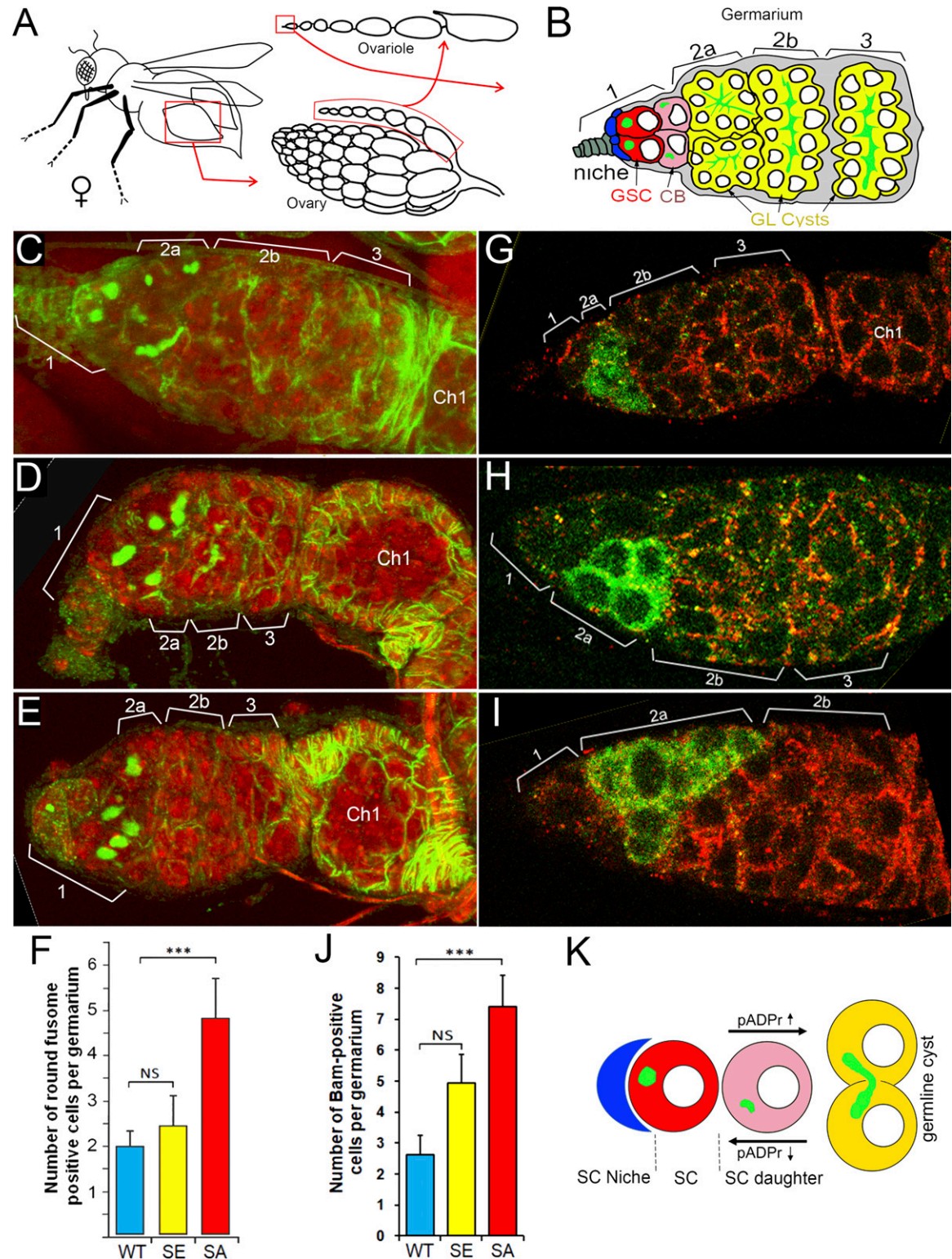

**Figure 4.** **Phosphorylation regulates poly(ADP-ribose) glycohydrolase (PARG) protein functions in *Drosophila* ovary germ-line stem cell (GSC) niche.**
**(A)** The structure of *Drosophila* ovary and ovariole. **(B)** Schematic illustration of anterior part of *Drosophila* ovariole and germanium. Cap cells (blue) form germline stem cell (GSC [red]) niche (area 1). Developing germline cells are shown in yellow (areas 2–3). Ch1, egg chamber stage 1. **(C, D, E, F)** Mutating sites of phosphorylation in PARG leads to overaccumulation of cells with a single fusome in the stem cell niche. **(D, E, F)** The organization of GSC niche is compared among WT PARG$^{WT}$ (D), phospho-mimetic PARG$^{SE}$ (E) and phospho-mutant PARG$^{SA}$ (F)-expressing *parg$^{27.1}$* animals. Green is a 1B1 antibody stain that marks Hu li tai shao protein, a component of the spectrosome, a round-shaped organelle specific to stem cells, and red is DNA. **(F)** Count of round fusome-positive cells per germarium, in *Drosophila* ovary based on four independent experiments. Blue is PARG$^{WT}$, yellow is PARG$^{SE}$, and red is PARG$^{SA}$. The statistical test is a two-tailed *t* test. ***P-value < 0.01, NS, nonsignificant. **(G, H, I, J)** Mutating sites of

balance between GSC maintenance and differentiation into CB. When we compared the effects of PARG[WT] (Fig S7C and D), PARG[SA] and PARG[SE] on *parg^{27.1}* mutants on germline stem cells maintenance and differentiation, we found that the phospho-mutant PARG isoform PARG[SA] shows a sharp increase in the number of round-shaped fusome-positive cells, up to five, on average (Fig 4E and F), whereas the *parg^{27.1}* mutant expressing either PARG[WT] or PARG[SE] shows no significant difference (Fig 4C, D, and F). To determine if these extra round-shaped fusome-positive cells are either GSCs or CBs, we checked the expression of Bam (Fig 4G–J). Interestingly, we found a significant increase in Bam-positive cells in PARG[SA] (Fig 4I and J) compared with PARG[WT] (Fig 4G and J), whereas PARG[SE] (Fig 4H and J) did not exhibit any significant increase. This suggests that the extra round-shaped fusome-positive cells observed in PARG[SA] are CBs, not GSCs. Taken together, these results suggest that PARG phosphorylation plays a regulatory role during CBs differentiation process (Fig 4K).

## Phosphorylation of PARG protein regulates egg-laying rate and is important for correct embryonic development

Next, we tried to determine if any defect in PARG phosphorylation would lead to a different female egg laying rate. Interestingly, PARG[SA] females lay a similar number of eggs compared with PARG[WT] (Fig S8). However, we noticed that PARG[SE] females significantly increased their egg production compared with PARG[WT]. A similar result was observed with the number of hatched eggs (Fig 5A), suggesting that the extra eggs laid by PARG[SE] finish their development. However, we observed that the proportion of unhatched eggs for PARG[SA] increase more than twofold over that of PARG[WT] or PARG[SE] (Fig 5B), corresponding to 29% of eggs expressing PARG[SA] that do not finish their development. Taken together, these results strongly suggest that PARG phosphorylation is important in regulating the egg-laying rate and correct embryonic development.

It was previously reported that egg production by *Drosophila* remains under strict control of food availability (38). Therefore, less food means less egg-laying production. To test if flies expressing PARG[SE] or PARG[SA] present any difference in egg-laying behavior compared with PARG[WT], five pairs of 1-d-old virgin flies were placed in vials containing molasses, agar and propionic acid with or without active yeast. Even without active yeast, this calorie-poor medium was sufficient to provide nutrients for flies. In normal condition, WT flies lay eggs on this medium only in the presence of active yeast (38). Therefore, females expressing only PARG[WT], PARG[SE] or PARG[SA] can still lay eggs, though few in number, on this calorie-poor medium in much the same way as that reported for WT flies (38).

We started by feeding the flies without active yeast (calorie-poor medium) and adding active yeast on Day 6 (calorie-rich medium).

In such conditions, WT flies do not lay eggs on calorie-poor medium, but rather start to lay eggs right after switching to a calorie-rich medium (38). Similar to WT flies, the females of our three conditions did not lay eggs on calorie-poor medium or start to lay eggs right after the switch to a calorie-rich medium. However, just after switching, we observed that females expressing PARG[SE] or PARG[SA] laid significantly fewer eggs than females expressing PARG[WT] (Fig 5C). 6 d after the switch, the PARG[SE] females still laid fewer eggs than control, but this difference did not turn out to be significant. PARG[SA] females, however, laid only half the number of the eggs laid by control (Fig 5C).

Then, we tried to start feeding flies with a calorie-rich medium before switching to a calorie-poor medium. In such conditions, WT flies stopped laying eggs right after the switch (38). The females of our three conditions started to lay eggs 1 d after they mated with males, as reported for WT flies (39). During the whole exposure to a calorie-rich medium (Day 1 to Day 6), PARG[SA] females laid around half the number of eggs compared with PARG[WT] and PARG[SE] females. This difference is significant. Just after the switch to a calorie-poor medium, the number of laid eggs drastically decreased for PARG[WT] and PARG[SA] females, whereas PARG[SE] females kept laying eggs at a rate similar to that before the switch (Fig 5D). Taken together, these results suggest that the dephosphorylation of PARG plays a role in coordinating the egg production process synchronized with the availability of a calorically sufficient food source.

## Phosphorylation of PARG regulates proper larval development and longevity

To further investigate the roles of PARG phosphorylation, we compared the "developmental speed rates" for PARG[WT], PARG[SE] or PARG[SA] expressing *parg^{27.1}* in synchronized populations. We measured the appearance of pupae (Fig 6A) and adult flies hatching (Fig 6B). On average, we observed that PARG[SA] pupae appeared with a delay of 1 d compared with PARG[SE] and PARG[WT] pupae. We also observed the same delay of 1 d in the appearance of PARG[SA] adults. These results suggest that PARG phosphorylation is important for the correct developmental timing of flies.

Next, we measured if the alteration of PARG phosphorylation sites would affect fly longevity. Similar to humans, WT *D. melanogaster* has a convex survivorship curve (40). Such curves are characterized by low mortality rates during early and middle life, but they rapidly increase after a certain age, 45 d post-pupation in the case of the WT *Drosophila* raised at 25°C (40). The PARG[WT] genotype in our study displayed a survivorship curve similar to what we would expect for WT flies. However, mutant genotypes rescued with PARG[SE] and PARG[SA], respectively, displayed a different survivorship curve shape. They were both characterized

phosphorylation in PARG leads to overaccumulation of Bag of marbles (Bam)-positive cells in the stem cell niche. **(G, H, I)** The organization of GSC niche is compared among WT PARG[WT] (G), phospho-mimetic PARG[SE] (H) and phospho-mutant PARG[SA] (I)-expressing *parg^{27.1}* animals. Green is a anti-BAM antibody stain that marks immediate daughters of GSC (cistoblasts). **(J)** Count of BAM-positive cells per germarium, in *Drosophila* ovary based on four independent experiments. Blue is PARG[WT], yellow is PARG[SE], and red is PARG[SA]. The statistical test is a two-tailed *t* test. \*\*\**P*-value < 0.01, NS, nonsignificant. **(K)** The model of the balance between stem cells (SCs) maintenance and differentiation of SC daughters. A higher level of pADPr leads to a shift of this balance in favor of differentiation, whereas a lower level of pADPr leads to a shift in favor of SC maintenance.

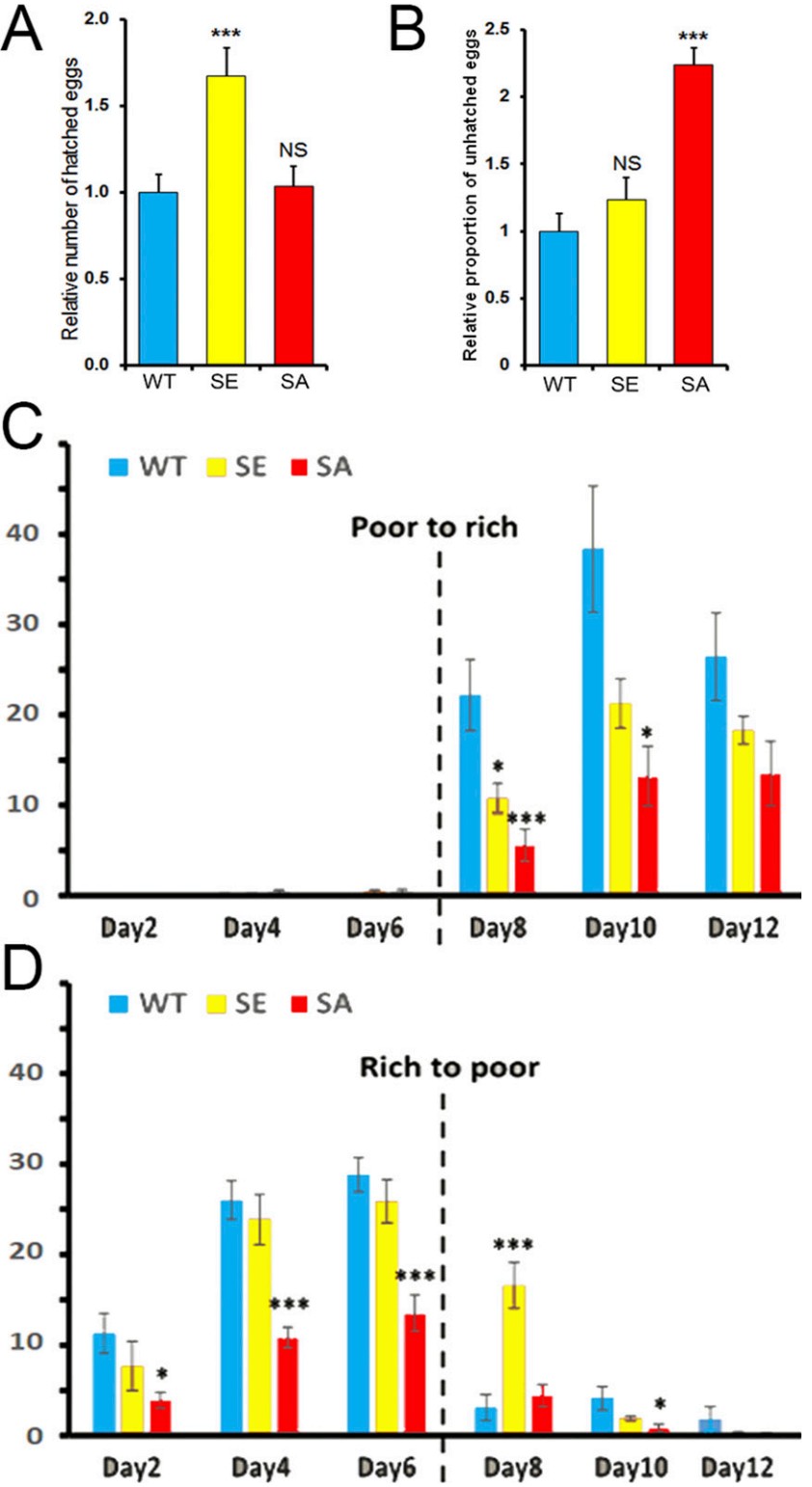

**Figure 5. poly(ADP-ribose) glycohydrolase phosphorylation defect leads to an increase in embryonic developmental arrest and affects the synchronization between egg production and food availability.**
**(A, B)** Quantification of the relative number of hatched (A) or unhatched (B) eggs laid by each female normalized by the average number of hatched (A) or unhatched (B) eggs laid by PARG$^{WT}$ females. These data are based on six independent experiments. **(C, D)** Number of eggs laid per female expressing PARG$^{WT}$ (blue), PARG$^{SE}$ (yellow), or PARG$^{SA}$ (red) per day in the presence of rich- or poor-calorie medium. The broken line corresponds to the switch between the presence and absence of active yeast. The number of eggs corresponds to the average of several vials (n = 3). The error bars represent the SEM. The statistical test realized is a two-tailed $t$ test (compared with PARG$^{WT}$). ***$P$-value < 0.01, *$P$-value < 0.05, NS, nonsignificant.

by a concave survivorship curve, with a high mortality rate, even during the first days of post-pupation (Fig 6C). Moreover, the survivability rate was affected in PARG$^{SA}$ and PARG$^{SE}$ flies compared with PARG$^{WT}$. Fifty percent of PARG$^{WT}$ flies were alive at Day 58, whereas 50% of PARG$^{SE}$ flies were alive at Day 42 and Day 30 for PARG$^{SA}$, corresponding to a difference of 28 d between PARG$^{SA}$ and PARG$^{WT}$. Data from males and females, when separated, presented results similar to those noted above (Fig 6D and E). To

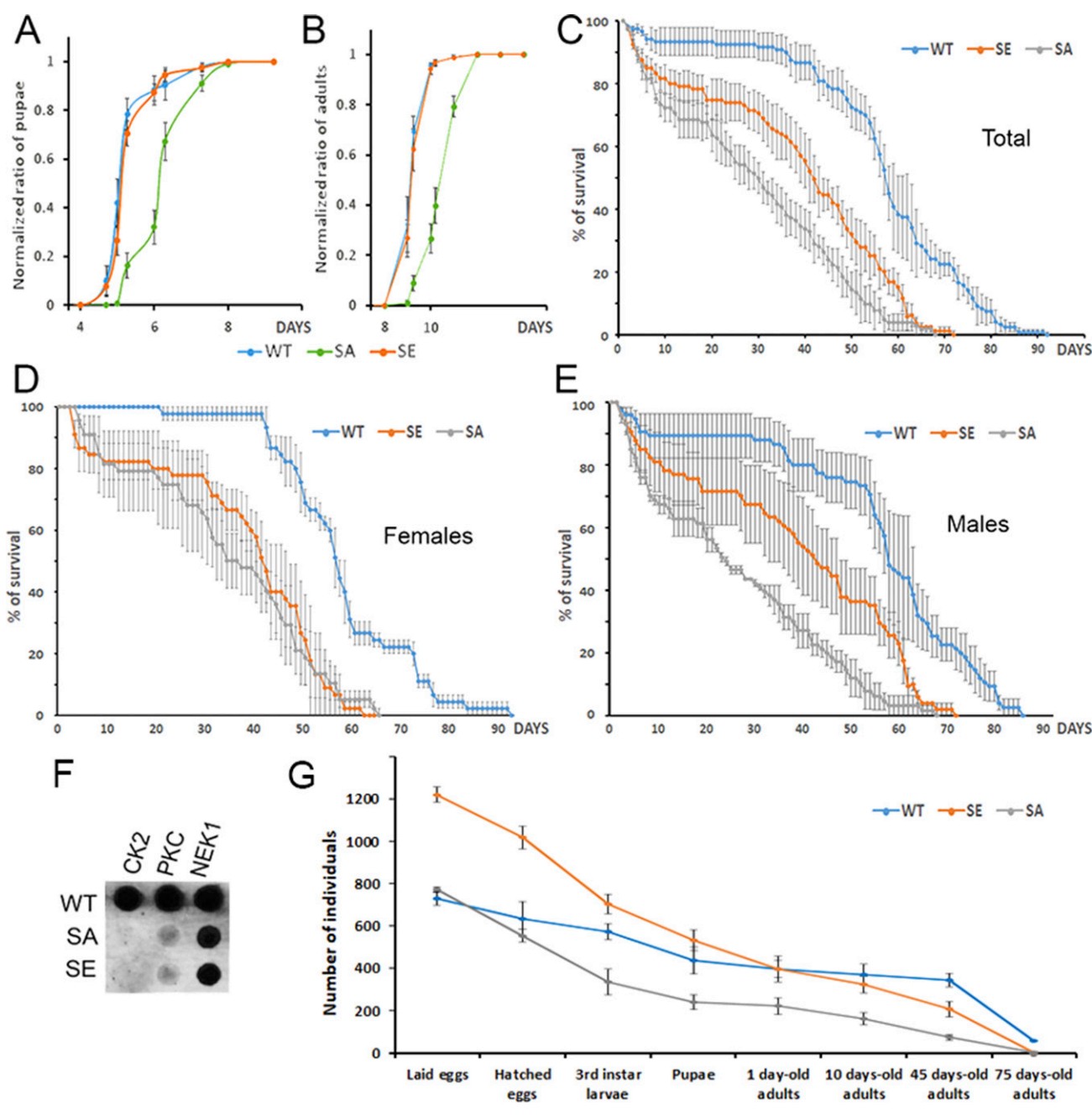

**Figure 6.   poly(ADP-ribose) glycohydrolase (PARG) phosphorylation regulates proper fly development and longevity.**
**(A)** Pupae appearance timing normalized by the total number of pupae in each bottle. This experiment was performed in six replicates. The error bar represents the SEM.
**(B)** Adult hatching timing normalized by the total number of adults in each bottle. This experiment was performed in six replicates. The error bar represents the SEM.
**(C, D, E)** Lifespan measurement of flies expressing PARG^WT (blue), PARG^SE (orange), or PARG^SA (grey). Y-axis represents the percentage of flies still alive on a specific Day (X-axis). Day 0 corresponds to adults hatching. **(C, D, E)** represents males and females mixed, whereas (D) and (E) represent females and males separated, respectively. Error bars represent SEM at each time point. The significance of the difference between curves was investigated using GLM analysis. The difference in survivorship is significant among PARG^SE, PARG^SA, and PARG^WT. **(F)** Mutating of PARG domains Ph1 and Ph2 abolishes PARG phosphorylation in a cell-free system by CK2 and PKC kinases, but not by NEK1. **(G)** Following of the progeny of 10 males and 30 females all along their life. PARG^WT are in blue, PARG^SE in yellow, and PARG^SA in red. The Y-axis corresponds to total of progeny we observed at each developmental stages. The data from "laid eggs" to 1-d-old adults" are directly observed, whereas the data from "10-d-old adults" to "45-d-old adults" are predicted based on our results presented Fig 6C–E.

test whether these survivorship curves are significantly different, we carried out GLM analysis with age in days as covariate and genotype as a grouping factor. We observed that PARG^SE and PARG^SA survivorship was significantly different from PARG^WT survivorship.

Taken together, these results suggest that PARG phosphorylation is important for *Drosophila* lifespan, but also for adult robustness because young PARG^SE and PARG^SA flies have a significantly shorter lifespan than PARG^WT flies.

## Phosphorylation domains of PARG are highly homologous to casein kinase 2 (CK2) and PKC motifs

To identify the enzyme responsible for PARG protein modification in vivo, we performed a motif analysis using NetPhorest ([41]). For all identified sites of phosphorylation, the software predicted the highest affinity for CK2 (Table S3). CK2 is known to be involved in several processes, such as cell signaling, embryogenesis and circadian clock ([42]). To test if CK2 could modify ph1 and ph2 in PARG protein, we produced recombinant PARG$^{WT}$, PARG$^{SE}$, and PARG$^{SA}$ isoforms of PARG using a bacterial system of protein purification ([43]). Besides CK2, we tested two other kinases also predicted by NetPhorest as candidates: PKC, known to be involved in cell polarity and cell asymmetric division in *Drosophila* ([44]), and NEK1. We purchased the commercially available enzymes and performed the kinase assay using ATP$_{32}$. We found that all three enzymes phosphorylate PARG$^{WT}$ in a cell-free system. However, mutating ph1 and ph2 abolishes activity only in CK2 and PKC, whereas NEK1 is still able to phosphorylate mutant PARG ([Fig 6F]) suggesting that only PKC and CK2 can phosphorylate PARG on ph1 and ph2. The only PARG sites reported to be phosphorylated in vivo in embryo and by our study are located on ph1 and ph2, suggesting that the phosphorylation of PARG observed in the presence of NEK1 only appears in a cell-free system, not in vivo. These results strongly suggest that both CK2 and PKC act in parallel to regulate PARG in *Drosophila*.

## Phosphorylation of PARG is required for proper development from embryos to adults

To have a better overview during which developmental stages PARG phosphorylation is important we combined the progeny analysis we did during eggs laying ([Fig S8]), eggs hatching ([Fig 5A]), third instar larvae appearance, pupae appearance ([Fig 6A]) and adult appearance ([Fig 6B]) with survivability rate we measured ([Fig 6C–E]). This allow us to visualize the progeny survivability all along the life or the individuals ([Fig 6G]). We found that only 29% of PARG$^{SA}$ progeny and 35% of PARG$^{SE}$ progeny survived from eggs to adults against 52% of PARG$^{WT}$. Overall, the survivability rate is lower in PARG$^{SE}$ and PARG$^{SA}$ than in PARG$^{WT}$ at all the developmental stages we checked except at pupal stage, which exhibit a similar ratio of third instar larvae that survive long enough to start pupation (84% for PARG$^{SA}$, 83% for PARG$^{SE}$, and 85% for PARG$^{WT}$). The more drastic difference we observed for PARG$^{SA}$ is from hatched eggs to third instar larvae where only 57% of PARG$^{SA}$ compared with 75% for PARG$^{WT}$, PARG$^{SE}$ exhibit an intermediate phenotype with 66% of survivability. Interestingly, the highest difference observed for PARG$^{SE}$ is from during emergence of adults where only 75% of PARG$^{SE}$ pupae survive long enough to generate adults flies compared with 84% for PARG$^{SA}$ and 91% for PARG$^{WT}$. Taken together, these results suggest that PARG correct phosphorylation is important all along the life of the flies, from eggs to adults.

# Discussion

The pADPr turnover has been studied for decades for its involvement in several critical functions, such as DNA repair, chromatin structure regulation, and transcriptional and translational activation and repression ([45], [46]). However, most studies are focused on PARP-1 regulation, not PARG regulation, which remains poorly understood. Several studies reported that human PARG can be phosphorylated at several sites ([Fig 7A]) ([22], [23], [24], [25], [26]) and that most of them are conserved in mice ([Fig S2]). All of these sites are predicted to be phosphorylated by PKC or by CKII kinases ([22], [24]). However, none of these phosphorylation sites is conserved in *Drosophila* ([Figs S2] and [7A]). Furthermore, the role of those phosphorylation sites in PARG activity is unknown. In this study, we confirmed six phosphorylation sites in *Drosophila* PARG that were identified at embryonic stages ([27]). We confirmed that those sites are also phosphorylated at larval stages ([Fig 2D]). All six sites are conserved among *Drosophila* species, but are absent from mammalian PARG ([Fig S2]). Interestingly, we did not detect a phosphorylation form of T$^{623}$ that has been reported to be phosphorylated in early embryo ([27]). Furthermore, this residue is not conserved among *Drosophila* species ([Fig S2]). This residue is not phosphorylated during third instar larvae but its phosphorylation might be important for PARG function during embryogenesis. We showed that alteration of those sites affected PARG protein quantity, but neither PARG protein localization nor parg mRNA quantity, suggesting that this difference occurs at the translational level or in PARG protein stability ([Figs 3] and [S4]). We also showed that the alteration of those sites decreases the adult lifespan. Finally, we showed strong evidence suggesting that PARG is phosphorylated by CKII or PKC kinases in a manner similar to that in PARG phosphorylation in mammals.

Interestingly, phosphorylation by CKII has been reported to protect phosphorylated proteins from degradation ([49]). Our results align with previously published data. We observed that the phospho-mutant PARG$^{SA}$ presents an accumulation of pADPr coupled with a lower protein level, which may suggest higher degradation compared to WT ([Fig 3B and C]). Furthermore, the phosphorylation-mimicking mutant PARG$^{SE}$ exhibits higher PARG protein quantity for the same level of mRNA compared to PARG$^{WT}$, which is compatible with the possibility that the PARG phosphorylated version is more stable. However, PARG$^{SE}$ exhibits a similar level of pADP hydrolysis compared with WT, despite a higher PARG protein quantity, suggesting that the phosphorylated version of PARG is less active than the non-phosphorylated version. Taken together, these results suggest that both phosphorylated and non-phosphorylated PARG are required for the correct function of PARG ([Fig 7B], top panel). The impossibility of switching from a phosphorylated state to a non-phosphorylated state seems to reduce PARG activity ([Fig 7B], middle panel). Furthermore, the impossibility of phosphorylating PARG decreases protein stability, leading to a drastic decrease in activity ([Fig 7B], lower panel). This drastic decrease in activity in PARG$^{SA}$ is enough to disrupt PARP-1 localization and diminish its amount in chromatin ([Fig 3D]). In this sense, PARG$^{SA}$ acts like a hypomorphic version of PARG with an intermediate phenotype, between mutant and control. However, the comparison between flies expressing PARG$^{SA}$ and parg$^{27.1}$ mutant flies is not possible because these two lines do not share the same genetic background. This difference of genetic background does not allow us to compare PARG$^{WT}$ with WT flies, as presented in [Fig 1D], as well. It is, however, possible that PARG$^{WT}$ presents some phenotypic differences compared with WT flies. These differences may result from

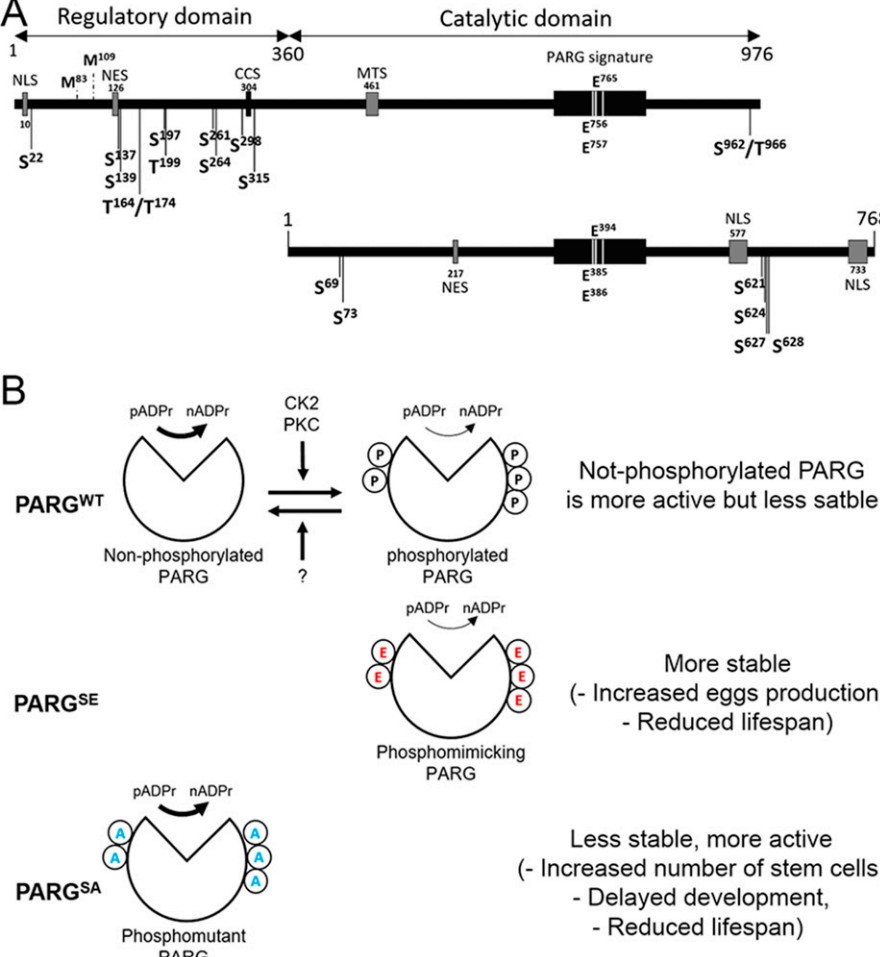

**Figure 7. Model of poly(ADP-ribose) glycohydrolase (PARG) protein activity regulation by phosphorylation.**
**(A)** Representation of PARG domains in Human (top panel) and in *Drosophila melanogaster* (bottom panel). PARG in Human includes a sequence 360 amino acids in length that represents a regulatory domain not conserved in *Drosophila*. Positions of the reported phosphorylation sites are highlighted with black arrows. S is serine and T is threonine. The sites separated by a slash correspond to sites, the residues of which are not confirmed between those possibilities. None of the phosphorylation sites reported in Human is conserved among *Drosophila* species. PARG *Drosophila* phosphorylation sites we reported in this study are conserved among *Drosophila* species, but not in mammals. Essential residues for catalytic activity are also highlighted. NLS, nuclear localization signal; NES, nuclear export signal; CCS, caspase cleavage site; MTS, mitochondrion transport signal. Mammalian NES (47) is not conserved among *Drosophila* species. The 217–223 *Drosophila* NES was predicted with NetNES (48). **(B)** Proposed model for the regulation of PARG activity. We postulate a state of homeostasis between a more stable phosphorylated PARG and more active non-phosphorylated PARG by CK2, PKC, and unidentified phosphatase (top panel). Both PARGs are needed for correct PARG activity. Because phospho-mimicking PARG[SE] cannot be dephosphorylated, this version is less sensitive to degradation, leading to an increased PARG pool compared to WT (middle panel). However, phosphorylated PARG is less active than non-phosphorylated PARG, leading to a similar pADPr hydrolysis rate than WT despite this increased PARG pool. Conversely because the phospho-mutant PARG[SA] version cannot be phosphorylated, this version is more sensitive to degradation, leading to a massive decrease in PARG pool compared with WT (bottom panel). The non-phosphorylated PARG version is more active than phosphorylated PARG. However, the decrease in PARG pool is enough to decrease PARG activity, leading to a significant decrease in pADPr hydrolysis rate compared with WT.

a combination of effects, including the difference in genetic background between these two lines and the possibility that PARG[WT]::YFP presents some minor differences from endogenous PARG.

We previously reported that the depletion of PARG in germarium affects the level of pADPr in GSC daughter cells leading to a complete loss of GSC maintenance (7). Here we reported that the replacement of PARG phosphorylated serine to alanine is responsible for an increase in pADPr level, leading to an accumulation of round-shape fusome and Bam-positive cells. The increase in both round-shape fusome and Bam-positive cells suggests a disruption of the differentiation of CBs into cysts rather than a disruption of the differentiation of GSC into CBs (Fig 4K).

We also observed an intriguing increase in embryonic developmental arrest, up to 29% of PARG[SA] eggs (more than twofold higher than that in PARG[WT]). We think that this defect was not reported in *parg*[27.1] mutants because *parg* mutant embryo inherits WT *parg* mRNA maternally, whereas PARG[SA] embryo inherits *parg*[SA] mRNA. We then concluded that this increase in unhatched eggs observed in PARG[SA] might reveal a role of PARG in embryonic development that is PARG phosphorylation dependent.

Furthermore, in the presence of dry yeast, PARG[SA] females lay a similar number of eggs compared with PARG[WT] (Fig S8), whereas in presence of active yeast, this number is significantly reduced compared with PARG[WT] (Fig 5C and D), suggesting that the egg-laying process of PARG[SA] females is more sensitive to nutrient availability compared with PARG[WT].

On the other hand, PARG[SE] females do not exhibit an increase in the number of round fusome-positive cells, whereas their egg production is increased compared with PARG[WT] females (Figs 4F and S8). A possible explanation is that PARG[SE] affects the speed of germline proliferation/differentiation, whereas PARG[SA] affects the differentiation program itself. Finally, PARG[SE] females fail to stop the egg-laying process in time during the switch from rich to poor medium, suggesting that the dephosphorylation of PARG is essential for the coordination between the egg production process and the availability of a calorically sufficient food source. It would be interesting to test if PARG phosphorylation status depends on the abundance of nutrients.

It is interesting to note that CK2 has been reported to phosphorylate oo18 RNA-binding protein (orb) in *Drosophila* ovaries (50). Orb phosphorylation is essential for oocyte specification, and

this disruption in phosphorylation leads to developmental arrest just after the 16-cell cysts stage, suggesting that CK2 is active in *Drosophila* ovaries before this stage. CK2 activity during oogenesis was reported in another study, highlighting the role of CK2 in the modulation of lipid metabolism during oogenesis by the phosphorylation of Jabba protein (51). The disruption of this phosphorylation leads to a decrease in female fertility. Taken together, these results suggest that CK2 activity is important along the entire egg production process that is subject to PARG phosphorylation. PKC is also activated in *Drosophila* germline in the establishment of the initial anterior-posterior polarity within cysts and in the maintenance of oocyte cell fate (52).

Furthermore, PARG^SA delays by 1 d the appearance of pupae and adult hatching (Fig 6A). Interestingly, we did not observe an increase in this delay during adult hatching. Therefore, in addition to embryonic developmental arrest, this suggests that larval development is delayed in the absence of phosphorylated PARG, but not pupal development. It is notable that the *parg^27.1* mutant does not exhibit this delay at larval stage (12). To explain, homozygote *parg* mutants die before pupation; therefore, we have to maintain a heterozygote stock. In this condition, every *parg* homozygote mutant larva received a maternal contribution from a heterozygote mother that possesses a WT copy of *parg*. Because PARG^SA rescues *parg* mutant lethality, the flies expressing PARG^SA can be homozygote for *parg^27.1* mutation. In that case, no maternal contribution is made with WT *parg*. Furthermore, this delay in larval development is also coupled with a lower survivability rate for PARG^SA larvae during larval development (Fig 6G), showing that PARG phosphorylation is important during larval development.

Finally, the impossibility of switching between phosphorylated and non-phosphorylated PARG affects adult lifespan (Fig 6). Lifespan is affected in two different ways. First, on average, flies that cannot switch between phosphorylated and non-phosphorylated PARG live at least 33% less than PARG^WT flies. Second, young adult flies are also less robust and die easily compared with PARG^WT. It would be interesting to test if this effect is strengthened under stress condition. This defect in adult lifespan is also coupled with a lower adults appearance ratio with only 75% of the pupae that live long enough to emerge as adults (Fig 6G). Furthermore, despite a higher number of laid eggs, the emergence of adults is similar in PARG^SE flies compared with PARG^WT, showing that all the extra progeny die during the eggs-adults period.

Overall, our data support that PARG phosphorylation is not only important in the regulation of GSC maintenance/differentiation into CB balance but all along the life of the flies, from embryonic development to adult longevity.

# Materials and Methods

### *Drosophila* strains and genetics

Genetic markers are described in FlyBase (53), and stocks were obtained from the Bloomington Stock Center, except as indicated. pP{w1, UAST::PARP-1-DsRed}, called UAS::PARP-1-DsRed, was described in reference 54. The transgenic stock with pP{w1, UAST::PARG-EYFP}, herein called PARG^WT, was described in reference 7.

The following GAL4 driver strains were used: 69B-GAL4 (54) and Arm::GAL4 (Bloomington stock no. 1560). Balancer chromosome carrying Kr::GFP, that is, FM7i, P{w1, Kr-GFP}, was used to identify heterozygous and homozygous *parg^27.1*.

### Construction of transgenic *Drosophila*

To make transgenic UAS::EYFP constructs containing mutant isoforms of PARG, we generated fragments of PARG cDNA. Primers used were as follows:

To mutagenize the phosphorylation site 1 Ser→Ala:

PARG-SA1-F1 - TGGCAATTGTCGAAGTGTGTGGTATTT
PARG-SA1-R1 - CCTT**CCATGG**AAACTCCACGCCACG**C**ATCATCTAGGGCGTTCG
PARG-SA1-F2 - CTG**CCATGG**AGGCTATACATCGTAATC
PARG-SA1-R2 - AGG**AGATCT**GCTGTTGGCTCAGGCC

To mutagenize the phosphorylation site 2 Ser→Ala:

PARG-SA2-F - AGC**TCTAGAG**TAGCTGGATTAGGCGAAGGAAAA**G**CA GAAACA**G**CAG\CGAAA**G**CC**G**CGCC
PARG-SA2-R - CTG**ACTAGT**GGTACCCTCGAGCCG

To mutagenize the phosphorylation site 1 Ser→Glu:

PARG-SE1-F - CCGGAAAATCTGGCGAACCAGCTAGATGAT**GAG**TGGCGTG**GA** GTTTCC
PARG-SE1-R - GGAAACTCCACGCCAC**TC**ATCATCTAG**CTC**GTTCGCCAGA TTTTCTGG

To mutagenize the phosphorylation site 2 Ser→Glu:

PARG-SE2-F - GGCGAAGGAAAA**GA**AGAAACA**GA**AGCGAAA**GAGGA**GCCA- GAACTCAACA AG
PARG-SE2-R - CTTGTTGAGTTCTGGC**TCCTC**TTTCGCT**TC**TGTTTCT**TC** TTTTCCTTCGCC

We used plasmid UAST::PARG-EYFP, containing full-length *Drosophila parg* cDNA clone, as a template for PCR amplification. The resulting PCR products were cloned directly into pUAST *Drosophila* vector in frame with EYFP using EcoRI and KpnI sites. *Drosophila* transformation was as described in reference 55, with modifications (56).

### Constructs of *Drosophila* PARG phospho-mutant and phospho-mimetic isoforms for protein purification

Full-length *Drosophila parg* and mutant isoforms S→A and S→E were inserted into expression vector pET-24(+), encoding a C-terminal 6-His-tag, and gene of bacterial kanamycin resistance. *Drosophila* transformation was as described in reference 55, with modifications (56). Flies expressing PARG^SE::YFP or PARG^SA:YFP were then crossed with *parg^27.1* flies to eliminate endogenous Parg expression.

### Purification/LC-MS/MS of SBP-protein complexes

Recombinant protein expression, affinity purification and detection. Rosetta DE3pLysS competent cells were transformed with each

respective recombinant plasmid and cultured on 0.5% glucose, kanamycin (50 μg/μl), and chloramphenicol (34 μg/ml) LB plates. A 10 ml aliquot of LB with glucose and respective antibiotics was inoculated with positive colonies and shaken overnight at 37°C. A 500 ml LB glucose/antibiotic solution was inoculated with the 10 ml sample and grown for ~2 h at 37°C. To induce expression, 5 ml of 100 mM isopropyl β-D-1-thiogalactopyranoside (IPTG) were added to the solution and incubated for 3 h at the same conditions. Purification was done using Ni column (GE Healthcare) and HPLC (GE Healthcare) according to the manufacturers' instructions. Detection of the respective proteins was performed after purification and Western blot assays using anti–His-tag antibody (ab9108; Abcam).

## Mass spectrometry analysis

Mass spectrometric identification of proteins was carried out as described in reference 32. Complete lanes from protein gels (Fig 2D) were cut into slices (narrow for specific bands) and analyzed by LC-MS/MS. The mass spectrometry (MS)/MS data were analyzed by nano-LC-MS/MS. Identified proteins were analyzed by the SAINT program. To identify PARG protein residues that are phosphorylated, slices gel corresponding to unmodified PARG and P-PARG* were used (Fig 2D). PARG protein phosphorylation was analyzed using nano-LC-MS/MS (Table S2).

## Western blot

The following antibodies were used for immunoblotting assays: anti-pADPr (Rabbit 1:4,000, #528815; Calbiochem), anti-pADPr (Mouse monoclonal, 1:500, 10H - sc-56198; Santa Cruz Biotechnology), anti-B-actin (Mouse monoclonal, 1:5,000, #A5441; Sigma-Aldrich), anti-Tubulin (Mouse monoclonal, 1:20,000, B512; Sigma-Aldrich) and anti-GFP (Mouse monoclonal, #632380, 1:4,000; BD Biosciences). Western blotting was performed using the detection kit from Amersham/GE Healthcare (#RPN2106), according to the manufacturer's instructions.

## *Drosophila* salivary gland polytene chromosome immunostaining

Preparation and immunostaining of polytene chromosome squashes were performed exactly as described (57). The primary antibody used was anti-GFP (Rabbit, #TP401, 1:400; Torrey Pines Biolabs), and the secondary antibody used was goat anti-rabbit Alexa-488 (Molecular Probes (1:1,500)). Slides were mounted in Vectashield (Vector Laboratories) with propidium iodide at 0.05 mg/ml for DNA staining.

## Quantitative RT-PCR assay

This assay was performed in triplicate. 12 third-instar larvae were collected for three groups (PARG$^{WT}$, PARG$^{SE}$, and PARG$^{SA}$). Total RNA was extracted from cells using the QIAshredder column and RNeasy kit (QIAGEN). Contaminating genomic DNA was removed by the g-column provided in the kit. cDNA was obtained by reverse transcription using M-NLV reverse transcriptase (Invitrogen). Real-time PCR assays were run using SYBR Green master mix (Bio-Rad) and an Applied Biosystems StepOnePlus

instrument. The amount of DNA was normalized using the difference in threshold cycle (CT) values (ΔCT) between *rpL32* and *parg* genes.

The quantitative real-time PCR (qPCR) primer sequences for *D. melanogaster* ribosomal protein L32 gene (*rpL32*) were 5'-GCTAAGCTGTCG-CAACAAAT-3' (forward) and 5'-GAACTTCTTGAATCCGGTGGG-3' (reverse).

Sequences for *parg* were 5'-AGAAACACCCTCAAGAGGAAG-3' (forward) and 5'-CGCTCTGTGGGACACAC-3' (reverse).

## Whole mount *Drosophila* tissue immunohistochemistry

Virgin females were collected and mated for 3 d before dissection. Ovaries dissected in Grace's insect medium were fixed in 4% PFA + 0.1% Triton X-100 in PBS for 20 min and blocked with 0.1% Triton X-100 + 1% BSA for 2 h. These ovaries were then incubated with mouse anti-Hu li tai shao antibody (1B1, 1:20; DSHB) or mouse anti-Bam (1:10; DSHB) overnight at 4°C, washed three times with PBS + 0.1% Triton X-100, and then incubated with fluorescence-labeled secondary antibody Alexa Fluor-488 goat anti-mouse (1:1,500; Invitrogen) for 2 h at room temperature. After washing three times with PBS + 0.1% Triton X-100, DNA in ovaries was stained with TOTO-3 Iodide (642/660) antibody (1:3,000, T3604; Thermo Fisher Scientific). Slides were mounted in Vectashield (Vector Laboratories).

## Egg-laying behavior

To test if flies that express PARG$^{SE}$ or PARG$^{SA}$ present any difference in egg-laying behavior compared with PARG$^{WT}$, five pairs of 1-d-old virgin flies were placed in vials containing molasses, agar and propionic acid with or without active yeast and covered with Kimwipe to prevent flies from sticking inside the food. This calorie-poor medium without active yeast provided enough nutrients for flies. In normal condition, WT flies lay eggs on this medium only in the presence of active yeast (38). This experiment was carried out in triplicate, and vials were changed daily at the same time. Eggs were counted just after transfer of flies. On day 6, the flies in vials containing yeast were transferred to vials without yeast (Rich to Poor), whereas flies in vials without yeast were transferred in vials containing yeast (poor to rich).

## Egg viability and developmental timing

To estimate how different constructs of PARG-YFP (PARG$^{WT}$, PARG$^{SE}$, or PARG$^{SA}$) affect the viability of eggs and the developmental timing of flies, 30 virgin females and 10 virgin males were collected for each condition and placed in a bottle containing regular *Drosophila* food and dry yeast. These flies were allowed to lay eggs for 3 h 30 min before transfer to another bottle for a total of six bottles. The number of eggs was counted just after transfer of the parents, whereas the proportion of unhatched eggs was calculated by counting the number of remaining eggs 4 d later. The number of pupae on the edge and at the surface of the food, as well as the number of adults, was counted at two time points every day. During the experiment, the parents and their progeny were kept at 25°C.

### Adult lifespan measurement

To measure if PARG phosphorylation impairment (PARG$^{SE}$ and PARG$^{SA}$) affects adult lifespan compared with PARG$^{WT}$, 25 virgin females and 15 virgin males for each condition were place in a tube containing standard cornmeal-molasses-agar media with dry yeast. The experiment was performed in triplicate. The flies were transferred to a fresh tube every 2 d, and deaths were counted daily. The flies were growth at 20°C.

### Kinase assay in vitro

Kinase assays were performed as described in reference 58. To detect phosphorylation of PARG protein isoforms, 1 μg of PARG$^{WT}$, PARG$^{SE}$ or PARG$^{SA}$ was mixed with 50–200 ng of CK2 (New England BioLabs, Inc.), PKC (Abcam), or NEK1 (SignalChem) kinases in kinase buffer (20 mM HEPES, pH 7.6, 1 mM MgCl$_2$, 1 mM EGTA, 5 mM NaF, 0.1 mM Na$_3$VO$_4$, 50 μM ATP, and 5 μM $_{p32}$ATP). Reaction mixtures were incubated at room temperature for 25 min, followed by loading 5mkl of reaction mixtures to the dot-blot nitrocellulose membrane. After the solution drying, the membrane was washed in TCA to remove free $_{p32}$ATP, and then signals were analyzed using autoradiography.

## Data Availability

Mutant strains and transgenic stocks are available upon request. The authors state that all data necessary to confirm the conclusions presented in the article are represented fully within the article.

## Supplementary Information

## Acknowledgements

Dr. Diane Darland contributed valuable comments on the manuscript. We also appreciate the assistance of students Michelle Currie, Jonathan Harbin, Michelle Ampofo, Haily Datz, Victor Gromoff, Breanna McLain, Cody Boyle, Brett MacLeod, Shri Patel, and Keely Walker in carrying out experiments, creating poly(ADP-ribose) glycohydrolase recombinant constructs, and imaging in vivo samples. Funding for this project was supported by the National Science Foundation MCB-1616740 and Department of Defense grant PC160049 to AV Tulin. Funding agencies had no role in study design, data collection, data analysis, interpretation, or writing of the report.

### Author Contributions

G Bordet: conceptualization, data curation, formal analysis, validation, investigation, methodology, and writing—original draft, review, and editing.
E Kotova: resources, data curation, and methodology.
AV Tulin: conceptualization, data curation, formal analysis, funding acquisition, validation, investigation, methodology, project administration, and writing—original draft, review, and editing.

### Conflict of Interest Statement

The authors declare that they have no conflict of interest.

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
