## [Reviewer comments · Life Science Alliance]

Life Science Alliance

Poly(ADP-ribosyl)ating pathway regulates development from stem cell niche to longevity control

Guillaume Bordet, Elena Kotova, and Alexei Tulin

DOI: <https://doi.org/10.26508/lsa.202101071>

Corresponding author(s): Alexei Tulin, University of North Dakota

Review Timeline:

Submission Date:	2021-03-17
Editorial Decision:	2021-05-19
Appeal Received:	2021-11-07
Editorial Decision:	2021-11-09
Revision Received:	2021-11-10
Editorial Decision:	2021-12-03
Revision Received:	2021-12-09
Accepted:	2021-12-10

Transaction Report:

May 19, 2021

Re: Life Science Alliance manuscript #LSA-2021-01071-T

Alexei Tulin
University of North Dakota

Dear Dr. Tulin,

Thank you for submitting your manuscript entitled "Poly(ADP-ribosyl)ation regulation controls development from eggs to aging". The manuscript has been evaluated by expert reviewers, whose reports are appended below. Unfortunately, after an assessment of the reviewer feedback, our editorial decision is against publication of the current version of this manuscript in Life Science Alliance.

We apologize for this unusual and extended delay in getting back to you. As you will note from the reviewers' comments below, both reviewers point out some similar concerns, i.e. poor description of methods, GSC counting, staging of oogenesis, and so on, while Rev 1 also points out confusion or ambiguity of many data presented, which must be clarified. In additional follow up, the reviewers agreed that the manuscript needs extensive revisions before it can be published in Life Science Alliance.

Although your manuscript is intriguing, we feel that the points raised by the reviewers are more substantial than can be addressed in a typical revision period. If you wish to expedite publication of the current data, it may be best to pursue publication at another journal.

Given the interest in the topic, we would be open to resubmission to Life Science Alliance of a significantly revised and extended manuscript that fully addresses the reviewers' concerns and is subject to further peer-review. If you would like to resubmit this work to Life Science Alliance, please submit an appeal directly through our manuscript submission system with a revised manuscript and a point-by-point rebuttal. Please note that priority and novelty would be reassessed at resubmission.

Regardless of how you choose to proceed, we hope that the comments below will prove constructive as your work progresses. We would be happy to discuss the reviewer comments further once you've had a chance to consider the points raised in this letter.

Thank you for thinking of Life Science Alliance as an appropriate place to publish your work.

Sincerely,

Shachi Bhatt, Ph.D.
Executive Editor
Life Science Alliance
<http://www.lsajournal.org>
Tweet @SciBhatt @LSAjournal

Reviewer #1 (Comments to the Authors (Required)):

The manuscript entitled as Poly(ADP-ribosyl)ation regulation controls development from egg to aging by Bordet et al addressed a function of phosphorylation of PARG during *Drosophila melanogaster* development and egg production. The comprehensive analysis revealed that PARG phosphorylation plays an important role in *Drosophila*. However, this manuscript is entirely poorly written. Some data are not properly described and/or interpreted. Data presentation is confusing. Most importantly, the manuscript provides no data suggesting a molecular mechanism for the function of PARG phosphorylation to explain their observations. In addition, a contradiction among some observations is not properly argued. I don't think this manuscript is ready for publication in its current format.

Major concern

In Fig1D and Fig3B, wt appears to be quite different. In addition, I wonder how much can PARG-SA rescue the defect in pADPr degradation. SA, SE, null and WT (if possible, EA as well) should be analyzed in the same blot.

In Fig3B, PARG(WT)-YFP which should be phosphorylated shows only a single band, but not any others. To confirm the phosphorylation of WT and the defects in SE and SA mutant forms, proteins must be analyzed for their phosphorylation using phos-tag gel or any other.

In Fig2B, PARG phosphorylation sites are shown in sequences of Fig2B without MS data. I wonder if T is not phosphorylated but all Ss are indeed phosphorylated. MS data should be provided.

In page 5 and Fig3D; "PARG-SA rescues PARP-1 localization only partially, and the amount of PARP-1 in chromatin is severely reduced". In Fig3D, however, PARP-1 appears to be localized onto chromatin, although the signal is weaker than that of the control. This is unclear whether the defect is due to chromatin localization or the weaker expression of PARP-1. To address this concern, WB data for PARP-1 should be provided.

In Fig4F, the authors claim that PARG-SA mutant germlium contained more GSC. I wonder if they are really GSCs, and what's behind. Is it due to perturbation of the niche signaling? In addition, how about differentiation defects? Phosphorylated Mad protein and the expression of Bam could be analyzed to address concerns above.

Fig4H and I; I do not really understand these panels. When unhatched eggs are increased, hatched eggs must be reduced, but SE and SA are not the case. The authors could simply provide the total egg number and hatching rate for each genotype. Do the authors claim that both SE and SA mutants lays more eggs (either hatching or unhatching)? These results seem to be contradictory to no change in the number of germ-line stem cells in Fig4G for SE. How do the authors interpret this?

In Fig4K, the author claims that "the phosphorylation of PARG plays a role in coordinating the egg production process synchronized with the availability of a calorically sufficient food source" However, SA mutant, albeit fewer egg laying, responds quite well to the nutrient change. How do the authors explain this inconsistency?

In Fig5, the wild-type control without any PARG must be included to exclude a possibility that PARG-wt may affect viability and/or longevity. In addition, any effect of the nutrient condition on the phosphorylation status of PARG must be analyzed.

Minor concern

The schematic drawings of the PARG are confusing. What are pink box and blue box representing a part of PARG? Similar colors (pink and red) should be avoided.

Page 5, "We observed that females expressing PARGSE significantly increased their production of eggs compared to PARG-WT, whereas PARG-SA females exhibited no significant difference (Fig.4H)." I think the authors meant 'hatching rate', not 'production of eggs'. 'egg viability' should be also 'hatching rate'. Similar mistakes of jargon can be found here and there.

In Page 25, Fig4 legend, what is egg chamber 1? Does it refer to stages? Stage 1 egg chamber?

The title of Fig3 should be changed to convey the authors' claim.

Fig3C, in Y-axis: PARG-GFP should be PARG-YFP. Some mistakes were also found in the text.

In Fig5, uniform the color of each genotype in longevity and survival plots.

Y-axis labels in Fig4G, J and K panels are missing.

Fig4I, "relative proportion of unfertilized eggs" in the figure, but it must be "proportion of unhatched eggs" as described in the text.

Are CK2 and PKC active in germline cells? And which stage?

There are many grammatical mistakes in sentences.

Reviewer #2 (Comments to the Authors (Required)):

In this manuscript, the author maps the two phosphorylation sites on Drosophila PRAG, which degrades poly(ADP-ribose) and thus remove poly(ADP-ribose) polymers from the targeted protein. The author also demonstrated PRAG phosphorylation is critical to maintain the proper levels of poly(ADP-ribose) polymers in the cells. The proper levels of poly(ADP-ribose) polymers are important for germline stem cell (GSC) maintenance, egg production in response to nutrition, development and lifespan. The results they show are clear, whereas less molecular mechanisms are provided.

Major points

1. None of the results is related to aging. The title of the manuscript needs to be modified.
2. Methodology for overexpression experiments is not fully described.

-Two GAL4 drivers were indicated in the Materials and Methods; however, it is not mentioned in what experiments these GAL4 line are used, where are GAL4 lines expressed, and what culture condition for performing knockdown experiments.

-The age of germaria they analyzed is not mentioned. It is also not clear how they expressed different constructs of PARG under *parg* mutant background.

3. The GSC counting is not accurate. The author obtained GSC number by counting the number of spectrosomes present in the germarium. However, the GSC fusome is not always in a round shape (spectrosome), it could be elongated. In addition, the cystoblast, the immediately daughter cell of GSCs, also carries a round shape fusome. Therefore, their GSC counting based on the number of spectrosomes is not accurate.

4. Since in their 2012 Nat. Comm. paper "Poly(ADP-ribose) controls DE-cadherin-dependent stem cell maintenance and oocyte localization", they had discovered that the association of pADPr and Hrp38, an orthologue of human hnRNPA1, disrupts the interaction of Hrp38 with the 5'-untranslated region of DE-cadherin messenger RNA, thereby diminishing DE-cadherin translation in progenitor cells.

Since *Parg(SA)* is more active to degrade pADPr. Is the increase of GSC number under *pargSA* overexpression/*parg* mutant background due to increased DE-cadherin expression levels in GSCs?

5. Lastly, can the author simply test the role of CK2 and PKCs in GSC maintenance, egg production, development and lifespan?
Minor points

1. The author should add explanation for Cajal bodies when it is shown first time in the text.

2. It is not clear why two germaria are shown in Fig. 4 B and C. What is the differences? Is it necessary to keep two?

3. The author should also briefly introduce the functions of CK2, PKC, and NEK1. In Fig. 6F, NEK1 should stronger phosphorylation on PARG mutant compared to wild-type. Can author explain this shortly?

4. The GSC fusome is not always in a round shape (spectrosome), it could be elongated. In addition, the cystoblast, the immediately daughter of GSCs also carries a round shape fusome. Therefore, their GSC counting based on the number of spectrosomes is not accurate.

5. The definition of germarial region 1-3 in this manuscript is different from other papers.

November 7, 2021

Dear Dr. Bhatt,

The authors of manuscript #LSA-2021-01071-T have requested an appeal. Their comments are below.

We thank the editors for the opportunity to resubmit our work and the reviewers as well for their careful reading of our manuscript and insightful suggestions. We significantly modified the manuscript according to reviewer's comments and highlight all the changed parts in yellow. We address all reviewers' comments point-by-point in the rebuttal letter and clarifies all those points in the revised text. We resubmit the revised paper as an article with 7 primary figures. We include a supplementary file, containing 2 supplemental tables and 8 supplemental figures. Thanks again for your consideration. Please do not hesitate to contact me if you need any further information. Thank you for the opportunity to resubmit our manuscript.

With best regards,
Alexei V. Tulin, Ph.D

You can accept or decline this request from the manuscript using the following link:

<https://lsa.msubmit.net/cgi-bin/main.plex?el=A3Na5bP4A6QqP6F6A9ftdLA23qCZZfdiu8Wg5w0aXUgZ>

Sincerely,

Editorial Staff

November 9, 2021

MS: LSA-2021-01071-T

Alexei Tulin
University of North Dakota

Dear Dr. Tulin,

Your revised manuscript entitled "Poly(ADP-ribosyl)ation regulation controls development from eggs to aging" has now been considered, and I am pleased to let you know that we have decided to send your manuscript back to the original Reviewers.

Please use the following link to submit your manuscript:

<https://lsa.msubmit.net/cgi-bin/main.plex?el=A2Na7bP4A3CkYN2I2B9ftdWWO03kERaiYuWJvcDBxHCwZ>

Yours sincerely,

Title

Poly(ADP-ribosyl)ating pathway regulates developmental processes from stem cell niche to longevity control

Dear Editor:

Thank you for the opportunity to resubmit our manuscript. We also thank the reviewers for their useful comments and suggestions to improve our manuscript. We resubmit the revised paper as an article with 7 primary figures. We include a supplementary file, containing 2 supplemental tables and 8 supplemental figures. This work is entirely novel and has not been published previously. In the enclosed document, we detail our responses to the comments of the reviewers.

As the manuscript was significantly rewritten and the line numbering changed, we highlighted all changed parts in yellow.

We thank the reviewers for their careful reading of our manuscript and insightful suggestions. The current revision addresses reviewers' comments point-by-point, and the revised text clarifies all of those points. Especially, we modified the Discussion to explain the molecular mechanisms underlying the regulation of developmental process by PARG protein, which was an issue of concern expressed by the reviewers.

We previously reported how the poly(ADP-ribosyl)ation pathway controls developmental processes in Drosophila GSC niche [Ji Y, Tulin AV. (2012)]. We showed that the absence of functional PARG in adult female germline leads to a complete loss of Germline stem cells (GSC) maintenance. In this revision, we discussed how new data fit this mechanism.

Despite this PARP-1/PARG pathway controls many molecular processes via regulation of transcription [Bordet G, Lodhi N, Guo D, Kossenkov A, Tulin AV. (2020)]. Thus, PARP-1/PARG system can act differently during different developmental stages and aging. Testing all these mechanisms is staying beyond the scope of this manuscript and including all these will create a monstrous manuscript.

Reviewers also suggested that single round-shaped fusome does not mark only germline stem cells in Drosophila germline. We agree that cystoblasts (CB), the direct GSC daughter cells that start their differentiation program exhibit a round-shaped fusome as well. Thus, we clarified this statement in the text and discuss this issue in the revised manuscript.

Although the reviewers suggested using anti-p-MAD antibody to test the change of "signaling" in GSC niche, this antibody is not commercially available, and multiple requests to share this antibody were either not answered or declined. To test how the disruption of PARG phosphorylation sites affect the maintenance-differentiation (M/D) balance we used anti-Bam Antibody. Bam-positive cells are immediate daughters of GSC that start their differentiation program. We found that mutating the phosphorylation domain of PARG protein significantly increases the amounts of Bam-positive cells. This observation fits totally with our previously published model [1]. The mutating of PARG phosphorylatable serine into alanine (PARGSA) leads to a decrease of PARG protein stability and behave as a hypomorphic version of PARG. The complete PARG knock-out in cell niche leads to an increase of GSC differentiation so important that it leads to a loss of GSC maintenance, while in PARGSA we have an attenuated phenotype with an increase of GSC differentiation that is not enough important to affect GSC maintenance.

Furthermore, our finding is not limited to niche cells. We showed that PARG phosphorylation is important all along development, from embryonic to larval development, but also for adult longevity. Moreover, the absence of correct regulation of PARG phosphorylation leads to an increase death ratio all along the life of the individuals and not only at adult stage.

We addressed every concern raised by reviewers in a specific manner.

Thanks again for your consideration. Please do not hesitate to contact me if you need any further information.

With best regards,

Alexei V. Tulin, Ph.D.

Point-by-point answers to reviewers' comments:

Reviewer #1:

The manuscript entitled as Poly(ADP-ribosyl)ation regulation controls development from egg to aging by Bordet et al addressed a function of phosphorylation of PARG during *Drosophila melanogaster* development and egg production. The comprehensive analysis revealed that PARG phosphorylation plays an important role in *Drosophila*. However, this manuscript is entirely poorly written. Some data are not properly described and/or interpreted. Data presentation is confusing. Most importantly, the manuscript provides no data suggesting a molecular mechanism for the function of PARG phosphorylation to explain their observations. In addition, a contradiction among some observations is not properly argued. I don't think this manuscript is ready for publication in its current format.

R1: Major concern:

R1: In Fig1D and Fig3B, wt appears to be quite different. In addition, I wonder how much can PARG-SA rescue the defect in pADPr degradation. SA, SE, null and WT (if possible, EA as well) should be analyzed in the same blot.

Response: The wild type in Figure 1D refers to yellow white flies that carry w1118 and y1 mutations. These flies are a good control to study *parg27.1* flies because these two lines share the same genetic background. In Figure 3B, flies expressing PARGWT-YFP, PARGSE-YFP or PARGSA-YFP can be compared as well since these three lines share the same genetic background. However, it is not possible to directly compare flies expressing PARGWT-YFP to yellow white flies since their genetic background is different. The comparison between *parg27.1* flies and PARGSA-YFP is not possible for the same reason. We modified our manuscript to clarify this statement.

R1: In Fig3B, PARG(WT)-YFP which should be phosphorylated shows only a single band, but not any others. To confirm the phosphorylation of WT and the defects in SE and SA mutant forms, proteins must be analyzed for their phosphorylation using phos-tag gel or any other.

Response: Discrimination between the phosphorylated and the non-phosphorylated band of PARG by Western blot needs a longer migration time owing to the small difference in their molecular weight. It is not possible to detect this double band with the migration condition we used for the Western blot presented Figure 3B. We added Supplemental Figure S7 that presents a Western blot with a longer migration time. Two bands can be observed in PARGWT-YFP, one at PARG-YFP expected size and one fainter at higher size, similar to our result presented Figure 2D. However, PARGSE-YFP and PARGSA-YFP only exhibit one band at expected size, suggesting that PARGSE-YFP and PARGSA-YFP cannot be phosphorylated anymore. We modified our manuscript to discuss this.

R1: In Fig2B, PARG phosphorylation sites are shown in sequences of Fig2B without MS data. I wonder if T is not phosphorylated but all Ss are indeed phosphorylated. MS data should be provided.

Response: We investigated all residues reported to be phosphorylated [Zhai et al., 2008] in *drosophila* PARG including T. Mutating T to A or E did not add any phenotypes to 6S-6A or 6S-6A. We did not include these results (as well results from mutating individual phosphorylation domains. These are very similar to wild type control). We reported only phosphorylation sites mutating which we modify PARG activity. Our MS data will be released upon the acceptance of the manuscript.

R1: In page 5 and Fig3D; "PARG-SA rescues PARP-1 localization only partially, and the amount of PARP-1 in chromatin is severely reduced". In Fig3D, however, PARP-1 appears to be localized onto chromatin, although the signal is weaker than

that of the control. This is unclear whether the defect is due to chromatin localization or the weaker expression of PARP-1. To address this concern, WB data for PARP-1 should be provided.

Response: We added Supplemental Figure S7 showing that PARP-1-DsRed protein level is similar in flies expressing PARGWT, PARGSE and PARGSA, suggesting that PARGSA does not affect PARP-1 protein level, but rather PARP-1 chromatin location. We modified the Results section to add this information.

R1: In Fig4F, the authors claim that PARG-SA mutant germarium contained more GSC. I wonder if they are really GSCs, and what's behind. Is it due to perturbation of the niche signaling? In addition, how about differentiation defects? Phosphorylated Mad protein and the expression of Bam could be analyzed to address concerns above.

Response: This is an interesting point. It is true that the cells with round-shaped fusome we observed in flies expressing PARGSA-YFP could be germline stem cells or cytotoblasts. To investigate this possibility, we stained germarium using Bam antibody, and we found that these extra cells with round-shaped fusome are, indeed, cytotoblasts, not germline stem cells. This could be attributed to the higher level of pADPr disrupting the balance between GSC maintenance and differentiation. We modified the Results section to clarify this statement.

R1: Fig4H and I; I do not really understand these panels. When unhatched eggs are increased, hatched eggs must be reduced, but SE and SA are not the case. The authors could simply provide the total egg number and hatching rate for each genotype. Do the authors claim that both SE and SA mutants lays more eggs (either hatching or unhatching)? These results seem to be contradictory to no change in the number of germ-line stem cells in Fig4G for SE. How do the authors interpret this?

Response: Figures 4H and I may be confusing. PARGSE does, indeed, lay a significantly higher number of eggs compared to wild type, whereas the difference between wild type and PARGSA is not significant. Regarding SE, one possible explanation for this increase would be a faster dynamic in the proliferation of GSC cells and the production of cyst cells, rather than a disturbance in the differentiation program. Regarding SA, we do see a small increase in egg production compared to wild type, but this difference is not significant. This is because the rate of unhatched eggs is low in wild type (around 13%), while the increased egg production observed in SA results from an increase of unhatched eggs only, leading to an increase of 124% of the number of unhatched eggs for an increase of 26% only in the egg production. This phenotype seems to be related to a role of PARG during embryonic development rather than a consequence of the disruption of the BGSC maintenance/ differentiation balance. We modified the Results and Discussion sections to discuss about this possibility, and we added Supplemental Figure S8 to address these points.

R1: In Fig4K, the author claims that "the phosphorylation of PARG plays a role in coordinating the egg production process synchronized with the availability of a calorically sufficient food source" However, SA mutant, albeit fewer egg laying, responds quite well to the nutrient change. How do the authors explain this inconsistency?

Response: In the experiment presented Figure 4K, we showed that PARGSE flies keep laying eggs at a similar rate after a switch to a calorie-poor medium. PARGSA response to this switch is similar to what we observed with PARGWT. This result suggests that the dephosphorylation of PARG is essential for a correct response to the switch from calorie-rich to calorie-poor medium. However, we noticed, as well, a significant difference between PARGSA flies that lay a number of eggs similar to that of PARGWT in the presence of dry yeast (Supplemental Fig. S8), whereas in the presence of active yeast, this amount is significantly decreased compared to PARGWT (Fig. 4J-K), suggesting that PARGSA egg-laying behavior is more sensitive to food environment compared to PARGWT. We modified the Discussion section to clarify this point.

R1: In Fig5, the wild-type control without any PARG must be included to exclude a possibility that PARG-wt may affect viability and/or longevity. In addition, any effect of the nutrient condition on the phosphorylation status of PARG must be analyzed.

Response: Our study focuses on the activity of phosphorylated PARG. PARGWT-YFP completely rescues lethality of parg27.1 mutant, and fly survivorship seems to be similar that of the wild-type survivorship curve. However, we do not claim that flies expressing PARGWT-YFP are identical to wild-type flies. We modified the Discussion section to clarify this point. The effects of nutrients on the phosphorylation status of PARG is an exciting avenue of investigation that may help to better characterize the role of PARG phosphorylation. However, this would require generating an antibody able to recognize the phosphorylated form of PARG. We tried several approaches to generate such antibody, but none succeeded. We modified the Discussion section to describe this experiment.

R1: Minor concern

R1: The schematic drawings of the PARG are confusing. What are pink box and blue box representing a part of PARG? Similar colors (pink and red) should be avoided.

Response: We modified Figure1F, 2A, 2C and 3A in the PARG schematic illustration to clarify PARG representation.

R1: Page 5, "We observed that females expressing PARGSE significantly increased their production of eggs compared to PARG-WT, whereas PARG-SA females exhibited no significant difference (Fig.4H)." I think the authors meant 'hatching rate', not 'production of eggs'. 'egg viability' should be also 'hatching rate'. Similar mistakes of jargon can be found here and there.

Response: We modified the manuscript to clarify the nomenclature.

R1: In Page 25, Fig4 legend, what is egg chamber 1? Does it refer to stages? Stage 1 egg chamber?

Response: Egg chamber 1 refers to Egg chamber stage 1. We modified the manuscript to clarify this statement.

R1: The title of Fig3 should be changed to convey the authors' claim.

Response: We modified the title of Figure 3.

R1: Fig3C, in Y-axis: PARG-GFP should be PARG-YFP. Some mistakes were also found in the text.

Response: We modified the manuscript to clarify the nomenclature.

R1: In Fig5, uniform the color of each genotype in longevity and survival plots.

Response: We modified the Figure 5 to match the color of each genotype between experiments.

R1: Y-axis labels in Fig4G, J and K panels are missing.

Response: We added Y-axis labels in Figure 4G, J and K.

R1: Fig4I, "relative proportion of unfertilized eggs" in the figure, but it must be "proportion of unhatched eggs" as described in the text.

Response: We modified the manuscript to clarify the nomenclature.

R1: Are CK2 and PKC active in germline cells? And which stage?

Response: CK2 has been reported to phosphorylate Orb protein in *Drosophila* ovaries (Wong et al., 2011). The decrease of Orb phosphorylation disrupts oocyte specification, leading to developmental arrest after the 16-cell cysts stage. This result suggests that CK2 is active in cyst cells before the 16-cell cyst stage. PKC is involved in the establishment of initial anterior-posterior polarity within cysts and in the maintenance of oocyte cell fate (Cox et al., 2001), suggesting that PKC is active in germline, at least the cyst stage. We modified the Results and Discussion section of our manuscript to include these statements.

R1: There are many grammatical mistakes in sentences.

Response: We fixed several grammatical mistakes in the manuscript.

Reviewer #2:

In this manuscript, the author maps the two phosphorylation sites on *Drosophila* PRAG, which degrades poly(ADP-ribose) and thus remove poly(ADP-ribose) polymers from the targeted protein. The author also demonstrated PRAG phosphorylation is critical to maintain the proper levels of poly(ADP-ribose) polymers in the cells. The proper levels of poly(ADP-ribose) polymers are important for germline stem cell (GSC) maintenance, egg production in response to nutrition, development and lifespan. The results they show are clear, whereas less molecular mechanisms are provided.

R2: Major points

R2: 1. None of the results is related to aging. The title of the manuscript needs to be modified.

Response: We changed the title to "Poly(ADP-ribosyl)ation regulation controls development from egg to longevity".

R2: 2. Methodology for overexpression experiments is not fully described. -Two GAL4 drivers were indicated in the Materials and Methods; however, it is not mentioned in what experiments these GAL4 line are used, where are GAL4 lines expressed, and what culture condition for performing knockdown experiments. -The age of germlaria they analyzed is not mentioned. It is also not clear how they expressed different constructs of PARG under *parg* mutant background.

Response: We modified the Materials and Methods section to clarify these points.

R2: 3. The GSC counting is not accurate. The author obtained GSC number by counting the number of spectrosomes present in the germarium. However, the GSC fusome is not always in a round shape (spectrosome), it could be elongated. In addition, the cystoblast, the immediately daughter cell of GSCs, also carries a round shape fusome. Therefore, their GSC counting based on the number of spectrosomes is not accurate.

Response: It is true that the cells with round-shaped fusome we observed in flies expressing PARGSA-YFP could be germline stem cells or cystoblasts. To investigate this possibility, we stained germarium using Bam antibody, and we found that these extra cells with round-shaped fusome are, indeed, cystoblasts, not germline stem cells. This could be attributed to the higher level of pADPr disrupting the balance between GSC maintenance and differentiation. We modified the Results and Discussion sections to clarify this statement.

R2: 4. Since in their 2012 Nat. Comm. paper "Poly(ADP-ribose) controls DE-cadherin-dependent stem cell maintenance and oocyte localization", they had discovered that the association of pADPr and Hrp38, an orthologue of human hnRNPA1, disrupts the interaction of Hrp38 with the 5'-untranslated region of DE-cadherin messenger RNA, thereby diminishing DE-cadherin translation in progenitor cells. Since Parg(SA) is more active to degrade pADPr. Is the increase of GSC number under pargSA overexpression/parg mutant background due to increased DE-cadherin expression levels in GSCs?

Response: This is an interesting point. We found that the extra round-shaped fusome-positive cells present in PARGSA germarium are CB and not GSC. This is consistent with the increase of pADPr quantity in PARGSA that should leads to more differentiation into CB. We discuss this result in the Discussion section.

R2: 5. Lastly, can the author simply test the role of CK2 and PKCs in GSC maintenance, egg production, development and lifespan?

Response: CK2 has been reported to phosphorylate Orb protein in *Drosophila* ovaries (Wong et al., 2011). The decrease of Orb phosphorylation disrupts oocyte specification, leading to developmental arrest after the 16-cell cyst stage. This result suggests that CK2 is active in cyst cells before the 16-cell cyst stage. Furthermore, CK2 has been reported to play an important role in the modulation of lipid metabolism during oogenesis and regulate female fertility (McMillan et al., 2018). PKC has been reported to be involved in the establishment of initial anterior-posterior polarity within cysts and in the maintenance of oocyte cell fate (Cox et al., 2001), suggesting that PKC is active in germline, at least at cyst stage. We modified the Discussion section to include these statements.

R2: Minor points

R2: 1. The author should add explanation for Cajal bodies when it is shown first time in the text.

Response: We modified the manuscript to present Cajal bodies in the Introduction section.

R2: 2. It is not clear why two germaria are shown in Fig. 4 B and C. What is the differences? Is it necessary to keep two?

Response: Figure 4B is a generic scheme to present the shape of germaria, whereas Figure 4C is a drawing of PARGWT-YFP germaria presented Figure 4D.

R2: 3. The author should also briefly introduce the functions of CK2, PKC, and NEK1. In Fig. 6F, NEK1 should stronger phosphorylation on PARG mutant compared to wild-type. Can author explain this shortly?

Response: We modified the Results section to introduce the functions of the different kinases. The experiment presented in Figure 6F was realized in a cell-free system, not *in vivo*. It shows that Nek1 is not able to phosphorylate PARG ph1 or ph2 sites, the only PARG phosphorylation sites reported in embryo and found with mass spectrometry. Therefore, the phosphorylation of PARGSE-YFP and PARGSA-YFP observed in the presence of NEK1 can only be observed *in vitro*. We modified the Results section of our manuscript to clarify this statement.

R2: 4. The GSC fusome is not always in a round shape (spectrosome), it could be elongated. In addition, the cystoblast, the immediately daughter of GSCs also carries a round shape fusome. Therefore, their GSC counting based on the number of spectrosomes is not accurate.

Response: It is true that cells with the round-shaped fusome we observed in flies expressing PARGSA-YFP could be germline stem cells or cystoblasts. To investigate this possibility, we stained germlarium using Bam antibody, and we found that these extra cells with round-shaped fusome are cystoblasts, not germline stem cells. We modified the Results and Discussion sections accordingly.

R2: 5. The definition of germlarium region 1-3 in this manuscript is different from other papers.

Response: We modified the definition of the germlarium region 1 to 3.

December 3, 2021

RE: Life Science Alliance Manuscript #LSA-2021-01071-TR-A

Prof. Alexei Tulin
University of North Dakota
501 North Columbia Road, Stop 90
GRAND FORKS, ND 582028367

Dear Dr. Tulin,

Thank you for submitting your revised manuscript entitled "Poly(ADP-ribosyl)ating pathway regulates development from stem cell niche to longevity control". We would be happy to publish your paper in Life Science Alliance pending final revisions necessary to meet our formatting guidelines. In your revision, please address Reviewer 1's remaining points.

- please upload your main manuscript text as an editable doc file
- please upload your Tables in editable .doc or excel format
- please add ORCID ID for the corresponding author-you should have received instructions on how to do so
- please add a Summary Blurb/Alternate Abstract in our system
- please add the Twitter handle of your host institute/organization as well as your own or/and one of the authors in our system
- Please upload all figure files as individual ones, including the supplementary figure files; all figure legends should only appear in the main manuscript file
- please add your main, supplementary figure, and table legends to the main manuscript text after the references section
- please note that titles in the system and manuscript file must match
- please add a conflict of interest statement to your main manuscript text
- please use the [10 author names, et al.] format in your references (i.e. limit the author names to the first 10)
- please add callouts for Figure S7A-D to your main manuscript text
- there is a callout for Figure 5F which does not exist, please correct

FIGURE CHECKS:

- Please add scale bars to Figures 1C and E, 3D, and indicate their size in the corresponding figure legend
- Please indicate molecular weight next to each protein blot
- The resolution of figures with western blots needs to be a minimum of 300 dpi.

A. FINAL FILES:

-- Summary blurb (enter in submission system): A short text summarizing in a single sentence the study (max. 200 characters including spaces). This text is used in conjunction with the titles of papers, hence should be informative and complementary to the title. It should describe the context and significance of the findings for a general readership; it should be written in the

present tense and refer to the work in the third person. Author names should not be mentioned.

B. MANUSCRIPT ORGANIZATION AND FORMATTING:

Sincerely,

Reviewer #1 (Comments to the Authors (Required)):

Overall, the authors provided, albeit not perfect, satisfactory responses. However, I found three issues: statement for GSC and CB, phosphorylation sites, and mass spec analysis. These issues should be further clarified to make this manuscript more valuable in the field.

First, regarding to GSC marker, pMad. It was surprising to find that the authors are not aware of the rabbit monoclonal antibody which has been widely used in the field. Many credible papers have successfully used the antibody which is commercially available (Smad3 (EP823Y) cross-reacts with fly pMad; e.g. DOI: 10.1016/j.devcel.2019.05.020). Nevertheless, the authors provided immunostaining with anti-Bam to differentiate GSCs and CBs, both of which contain round fusome, and showed that CB expressing Bam, but not GSCs, are increased in PARG[SA]. These results suggest that differentiation into CB from GSC appears unaffected, but rather, differentiation (or mitotic division) of CB into cystocytes appears affected there. Hence, the statement below, the balance between GSC maintenance and differentiation leads misinterpretation.

Page 6, "Taken together, these results suggest that PARG phosphorylation plays a regulatory role in balancing GSC maintenance and differentiation (Fig.4K)."

Page 11, "leading to disruption of the balance between stem cell maintenance and differentiation"

The authors should provide further careful argument to avoid readers' confusion.

The authors did not describe clearly the MS analysis for the phosphorylated peptides. Fig2B showed the sequences of PARG phospho-peptides identified in embryos in the previous study. "S" or "T" in red appear to be phosphorylated residues proposed in the previous study, but it is not clear if so or not.

The authors also performed the mass analysis for PARG protein and confirmed the two out of five peptides shown in Fig2B. However it is not shown what they confirmed. They did not provide the phosphorylated sites in their mass analysis. Thus, it is

not clear if the authors detect the phosphorylated peptides or no-modified peptides.

Importantly, the previous mass analysis proposed 5 possible phosphorylation sites in Ph2 region (four Ser residues and one Thr residue). The authors focused on four Ser residues in their analysis without mentioning any reason to exclude Thr contribution.

The authors mentioned in their response that the mutation of T did not affect the PARG activity in the 6S PARG mutant context. This observation can be added in the manuscript. However, it is still possible that the phosphorylation on Thr may be equivalent to that on Ser in any other context.

Reviewer #2 (Comments to the Authors (Required)):

The author has fully addressed my concerns, and I have no further comments.

RE: "Poly(ADP-ribosyl)ating pathway regulates development from stem cell niche to longevity control"

Dear Editors:

Thank you for considering our paper for publication in *LSA*. We are grateful to the reviewers for the insightful comments and revised the manuscript in accord with their suggestions. We have modified our manuscript to clarify the identification of Parg phosphorylation sites during third instar larvae and the role of Parg in germline stem cells maintenance/differentiation processes. We also included Supplemental Table S2 to present the different peptides combination that was found during mass-spectrometry analysis.

Please contact me if you need any further information.

With best regards,

Alexei Tulin

Reviewer 1:

R1: Overall, the authors provided, albeit not perfect, satisfactory responses. However, I found three issues: statement for GSC and CB, phosphorylation sites, and mass spec analysis. These issues should be further clarified to make this manuscript more valuable in the field.

Response: We thanks Reviewer 1 for his careful reading of our manuscript and his useful insights he provided.

R1: First, regarding to GSC marker, pMad. It was surprising to find that the authors are not aware of the rabbit monoclonal antibody which has been widely used in the field. Many credible papers have successfully used the antibody which is commercially available (Smad3 (EP823Y) cross-reacts with fly pMad; e.g. DOI: 10.1016/j.devcel.2019.05.020).

Response: We contacted several of our collaborators and none of them were able to share pMad antibody with us. We were indeed not aware about this antibody and we thanks Reviewer 1 for this information. We will order this antibody for future experiments.

R1: Nevertheless, the authors provided immunostaining with anti-Bam to differentiate GSCs and CBs, both of which contain round fusome, and showed that CB expressing Bam, but not GSCs, are increased in PARG[SA]. These results suggest that differentiation into CB from GSC appears unaffected, but rather, differentiation (or mitotic division) of CB into cystocytes appears affected there. Hence, the statement below, the balance between GSC maintenance and differentiation leads misinterpretation.

R1: Page 6, "Taken together, these results suggest that PARG phosphorylation plays a regulatory role in balancing GSC maintenance and differentiation (Fig.4K)." Page 11, "leading to disruption of the balance between stem cell maintenance and differentiation". The authors should provide further careful argument to avoid readers' confusion.

Response: We modified results, discussion part and the model presented Figure 4K to clarify this point and avoid confusion about these data.

R1: The authors did not describe clearly the MS analysis for the phosphorylated peptides. Fig2B showed the sequences of PARG phospho-peptides identified in embryos in the previous study. "S" or "T" in red appear to be phosphorylated residues proposed in the previous study, but it is not clear if so or not. The authors also performed the mass analysis for PARG protein and confirmed the two out of five peptides shown in Fig2B. However it is not shown what they confirmed. They did not provide the phosphorylated sites in their mass analysis. Thus, it is not clear if the authors detect the phosphorylated peptides or no-modified peptides. Importantly, the previous mass analysis proposed 5 possible phosphorylation sites in Ph2 region (four Ser residues and one Thr residue). The authors focused on four Ser residues in their analysis without mentioning any reason to exclude Thr contribution. The authors mentioned in their response that the mutation of T did not affect the PARG activity in the 6S PARG mutant context. This observation can be added in the manuscript. However, it is still possible that the phosphorylation on Thr may be equivalent to that on Ser in any other context.

Response: To identify PARG protein residues that are phosphorylated, the gel presented figure 2D was cut between P-PARG* and PARG bands. Then, the peptides present in the upper and lower part of the gel were analyzed by mass spec separately. We did not detect phosphorylation of T⁶²³ residue that was previously reported in early embryo. We modified Material & Methods section to clarify this. We also added new **Supplemental Table S2**, which clearly demonstrates that T⁶²³ residue is never phosphorylated in our approaches.

We identified the phosphorylation sites at third instar larvae, suggesting that this T⁶²³ residue is not phosphorylated during third instar larvae. Moreover, taking into account that by mutating T⁶²³ we did not detect any phenotypes in present experiments. We generated and tested 17 more transgenic constructs, with different combinations of T and S mutations and mimicking phosphorylation. We never detect an effect of the absence of this T⁶²³ residue in any condition. Including all of these data will be too heavy addition with not bringing additional information. We preferred to avoid "overloading" our report with "no effects" data. Furthermore, we found that the 6 Serines residues are conserved among *Drosophila* species while T⁶²³ is not (**supplemental figure 2**).

However, we agree that phosphorylation of T⁶²³ may play roles in some specific cell times or in specific condition, likely during embryonic stage so we included this statement to the discussion.

Reviewer 2:

The author has fully addressed my concerns, and I have no further comments.

Response: We thanks Reviewer 2 for his careful reading of our manuscript and his useful insights he provided.

December 10, 2021

RE: Life Science Alliance Manuscript #LSA-2021-01071-TRR

Prof. Alexei Tulin
University of North Dakota
501 North Columbia Road, Stop 90
GRAND FORKS, ND 582028367

Dear Dr. Tulin,

Thank you for submitting your Research Article entitled "Poly(ADP-ribosyl)ating pathway regulates development from stem cell niche to longevity control". It is a pleasure to let you know that your manuscript is now accepted for publication in Life Science Alliance. Congratulations on this interesting work.

DISTRIBUTION OF MATERIALS:

Again, congratulations on a very nice paper. I hope you found the review process to be constructive and are pleased with how the manuscript was handled editorially. We look forward to future exciting submissions from your lab.

Sincerely,
